# Hematopoietic stem and progenitor cells regulate the regeneration of their niche by secreting Angiopoietin-1

Bo O Zhou[1†], Lei Ding[1,2,3*†], Sean J Morrison[1*]

[1]Department of Pediatrics and Children's Research Institute, Howard Hughes Medical Institute, University of Texas Southwestern Medical Center, Dallas, United States; [2]Department of Rehabilitation and Regenerative Medicine, Columbia Stem Cell Initiative, Columbia University Medical Center, New York, United States; [3]Department of Microbiology and Immunology, Columbia Stem Cell Initiative, Columbia University Medical Center, New York, United States

**Abstract** Hematopoietic stem cells (HSCs) are maintained by a perivascular niche in bone marrow but it is unclear whether the niche is reciprocally regulated by HSCs. Here, we systematically assessed the expression and function of *Angiopoietin-1* (*Angpt1*) in bone marrow. *Angpt1* was not expressed by osteoblasts. *Angpt1* was most highly expressed by HSCs, and at lower levels by c-kit[+] hematopoietic progenitors, megakaryocytes, and Leptin Receptor[+] (LepR[+]) stromal cells. Global conditional deletion of *Angpt1*, or deletion from osteoblasts, LepR[+] cells, *Nes-cre*-expressing cells, megakaryocytes, endothelial cells or hematopoietic cells in normal mice did not affect hematopoiesis, HSC maintenance, or HSC quiescence. Deletion of *Angpt1* from hematopoietic cells and LepR[+] cells had little effect on vasculature or HSC frequency under steady-state conditions but accelerated vascular and hematopoietic recovery after irradiation while increasing vascular leakiness. Hematopoietic stem/progenitor cells and LepR[+] stromal cells regulate niche regeneration by secreting Angpt1, reducing vascular leakiness but slowing niche recovery.

*For correspondence: ld2567@ columbia.edu (LD); sean. morrison@utsouthwestern.edu (SJM)

†These authors contributed equally to this work

## Introduction

Hematopoietic stem cells (HSCs) reside in a specialized bone marrow niche in which Leptin Receptor[+] (LepR[+]) perivascular stromal cells and endothelial cells secrete factors that promote their maintenance (*Kobayashi et al., 2010*; *Ding et al., 2012*; *Ding and Morrison, 2013*; *Greenbaum et al., 2013*; *Poulos et al., 2013*; *Morrison and Scadden, 2014*). Nearly all the cells that express high levels of *Scf* (*Kitl*) or *Cxcl12* in the bone marrow are LepR[+] (*Zhou et al., 2014*). Conditional deletion of *Scf* from LepR[+] cells and endothelial cells leads to loss of all quiescent and serially-transplantable HSCs from adult bone marrow (*Oguro et al., 2013*). These LepR[+] niche cells have also been identified based on their expression of high levels of *Cxcl12* (*Sugiyama et al., 2006*; *Ding and Morrison, 2013*; *Omatsu et al., 2014*), low levels of the *Nestin*-GFP transgene (*Mendez-Ferrer et al., 2010*; *Kunisaki et al., 2013*), PDGFRα (*Morikawa et al., 2009*; *Zhou et al., 2014*), and *Prx1*-Cre (also known as *Prrx1*-Cre) (*Greenbaum et al., 2013*). Consistent with the conclusion that HSC niche cells include mesenchymal stem/stromal cells (*Sacchetti et al., 2007*; *Mendez-Ferrer et al., 2010*), the LepR[+] cells are highly enriched for CFU-F and give rise to most of the osteoblasts and fat cells that form in adult bone marrow (*Zhou et al., 2014*).

*Angpt1* has been proposed to be expressed by osteoblasts in the bone marrow and to promote the maintenance of quiescent HSCs in an osteoblastic niche (*Arai et al., 2004*). However, HSCs and perivascular stromal cells also express *Angpt1* (*Takakura et al., 2000*; *Ivanova et al., 2002*;

**eLife digest** In adults, blood cells develop from a set of stem cells that are found in bone marrow. There are also specialized blood vessels and cells called 'stromal cells' within the bone marrow that provide these stem cells with oxygen, nutrients, and other molecules. This local environment, or 'niche', plays an important role in regulating the maintenance of these stem cells. But it has not been known whether stem cells can reciprocally regulate their niches.

Unfortunately, radiation used to treat cancer obliterates the stem cells and their niche; both must recover after such a treatment before the patient can produce blood cells normally again. A protein called Angpt1 is thought to play a role in this post-treatment recovery. Angpt1 is known to regulate blood vessels in the bone marrow, and one influential study had previously suggested that bone cells produce Angpt1, which promotes and regulates the maintenance of the stem cells within the niche. However, this previous study did not directly test this. Thus, it was not clear whether Angpt1 promotes the regeneration of the stem cells themselves or if it regulates the rebuilding of the niche.

Now, Zhou, Ding and Morrison have genetically engineered mice to make a 'reporter' molecule—which glows green when viewed under a microscope—wherever and whenever the gene for Angpt1 is active. These experiments showed where the protein is produced, and unexpectedly revealed that the bone cells do not make Angpt1. Instead, it is the stem cells and the stromal cells in the niche that made the protein. Further experiments showed that deleting the gene for Angpt1 from mice, or just from their bone cells, did not affect blood cell production; nor did it affect the maintenance or regulation of the stem cells.

Next, Zhou, Ding and Morrison looked at whether Angpt1 might be involved in rebuilding the niche after being exposed to radiation. Some of these irradiated mice had been genetically engineered to lack Angpt1; and, in these mice, blood stem cells and blood cell production recovered more quickly than in mice with Angpt1. The blood vessels in the niche also grew back more quickly in the irradiated mice that lacked Angpt1. However, these regenerated blood vessels were leaky. This suggests that blood stem cells produce Angpt1 to slow the recovery of the niche and reduce leakage from the blood vessels. Thus, blood stem cells can regulate the regeneration of the niches that maintain them.

---

*Forsberg et al., 2005*; *Kiel et al., 2005*; *Sacchetti et al., 2007*; *Ding et al., 2012*). Moreover, it has not been tested whether *Angpt1* deficiency affects HSC function in vivo. Thus, the physiological function and sources of Angpt1 in the bone marrow remain uncertain.

Angpt1 (*Suri et al., 1996*), and its receptor Tie2 (*Dumont et al., 1994*; *Puri et al., 1995*; *Sato et al., 1995*; *Davis et al., 1996*), are necessary for embryonic vascular development. Tie2 is mainly expressed by endothelial cells (*Schnurch and Risau, 1993*; *Kopp et al., 2005*) but also by HSCs (*Iwama et al., 1993*; *Arai et al., 2004*). *Angpt1* over-expression promotes the development of larger, more numerous, more highly branched, and less leaky blood vessels (*Suri et al., 1998*; *Thurston et al., 1999*; *Cho et al., 2005*). *Angpt1* expression by primitive hematopoietic progenitors (HPCs) promotes angiogenesis during embryonic development (*Takakura et al., 2000*). Global conditional deletion of *Angpt1* between embryonic day (E)10.5 and E12.5 increases the size and number of blood vessels in fetal tissues but later deletion has little effect on vascular development (*Jeansson et al., 2011*). Nonetheless, Angpt1 does regulate angiogenesis in response to a variety of injuries in adult tissues (*Kopp et al., 2005*; *Jeansson et al., 2011*; *Lee et al., 2013*), promoting angiogenesis in some contexts (*Thurston et al., 1999*) while negatively regulating angiogenesis in other contexts (*Visconti et al., 2002*; *Augustin et al., 2009*; *Jeansson et al., 2011*; *Lee et al., 2014*). A key function of Angpt1 is to reduce the leakiness of blood vessels, perhaps by tightening junctions between endothelial cells (*Thurston et al., 1999*; *Brindle et al., 2006*; *Lee et al., 2013*, *2014*).

Irradiation and chemotherapy not only deplete HSCs but also disrupt their niche in the bone marrow, particularly the sinusoids (*Knospe et al., 1966*; *Kopp et al., 2005*; *Li et al., 2008*; *Hooper et al., 2009*) around which most HSCs (*Kiel et al., 2005*) as well as *Scf-*, *Cxcl12-*, and LepR-expressing stromal cells reside (*Ding et al., 2012*; *Ding and Morrison, 2013*; *Omatsu et al., 2014*; *Zhou et al., 2014*). Regeneration of this perivascular niche after injury, including endothelial and stromal components, is necessary for regeneration of HSCs and hematopoiesis (*Kopp et al., 2005*; *Hooper*

*et al., 2009*). After 5-fluorouracil treatment, Tie2 signaling (which is regulated by its ligands Angpt1, Angpt2, and possibly Angpt3 [*Augustin et al., 2009*; *Eklund and Saharinen, 2013*; *Fagiani and Christofori, 2013*; *Thomson et al., 2014*]) regulates the remodeling of blood vessels in the bone marrow and adenoviral over-expression of *Angpt1* accelerates the recovery of hematopoiesis (*Kopp et al., 2005*). This raises the question of whether endogenous *Angpt1* is necessary for niche recovery and whether it acts by promoting HSC function in an osteoblastic niche or by regulating vascular regeneration.

## Results

### *Angpt1* is expressed by megakaryocytes, HSCs, c-kit$^+$ cells, and LepR$^+$ stromal cells

We first assessed the Angpt1 expression using a commercially available antibody to stain bone marrow sections. Most bone marrow cells did not stain positively and we were unable to detect any staining among bone-lining cells where osteoblasts localize (*Figure 1A–C*). The most prominent staining was in large CD41$^+$ megakaryocytes (*Figure 1D–F*) and in c-kit$^+$ HPCs (*Figure 1G–I*).

To analyze *Angpt1* expression by flow cytometry, we generated *Angpt1$^{GFP}$* knock-in mice by recombining *GFP* into the endogenous *Angpt1* locus (*Figure 1—figure supplement 1A–D*). Consistent with the antibody staining pattern, GFP was expressed by CD41$^+$ megakaryocytes (*Figure 1J–L*) and c-kit$^+$ HPCs throughout bone marrow (*Figure 1M–O*). By flow cytometry, only $1.5 \pm 0.8\%$ of mechanically dissociated bone marrow cells (which include few stromal cells) were GFP$^+$ (*Figure 1P*). Overall, 85% of GFP$^+$ hematopoietic cells were c-kit$^+$ (*Figure 1—figure supplement 1E*): $72 \pm 13\%$ of c-kit$^+$ cells were GFP$^+$ and only $1.3 \pm 0.7\%$ of c-kit$^-$ cells were GFP$^+$ (*Figure 1Q,R*). All CD150$^+$CD48$^-$LSK HSCs expressed high levels of GFP (*Figure 1S*). All CD150$^-$CD48$^-$LSK multipotent progenitors (MPPs) were also positive for GFP, though at somewhat lower levels per cell than HSCs (*Figure 1T*). Virtually all CD48$^+$LSK HPCs, Lineage$^-$Sca1$^{low}$c-kit$^{low}$Flt3$^+$IL7Rα$^+$ common lymphoid progenitors (CLPs; *Kondo et al., 1997*), CD34$^+$FcγR$^-$Lineage$^-$Sca1$^-$c-kit$^+$ common myeloid progenitors (CMPs; *Akashi et al., 2000*), and CD34$^+$FcγR$^+$Lineage$^-$Sca1$^-$c-kit$^+$ granulocyte-monocyte progenitors (GMPs; *Akashi et al., 2000*) were GFP$^+$ but with successively lower expression levels per cell relative to HSCs (*Figure 1U–X*). Few CD34$^-$FcγR$^-$Lineage$^-$Sca1$^-$c-kit$^+$ megakaryocyte-erythroid progenitors (MEPs; *Akashi et al., 2000*) were positive for GFP (*Figure 1Y*). *Angpt1* was thus broadly expressed by early HPCs, at levels that declined as progenitors matured.

Virtually all of the *Angpt1* expression by stromal cells in the bone marrow was by LepR$^+$ cells. GFP$^+$ LepR$^+$ stromal cells localized mainly around sinusoids throughout the bone marrow (*Figure 2A–C*) but were also present near arterioles (data not shown). GFP was expressed by $94 \pm 3.2\%$ of LepR$^+$ stromal cells and $94 \pm 2.5\%$ of GFP$^+$ stromal cells (CD45$^-$Ter119$^-$) were LepR$^+$ (*Figure 2F*). Consistent with this, nearly all GFP$^+$ stromal cells were PDGFRα$^+$, a marker of mesenchymal stem/stromal cells (*Morikawa et al., 2009*) expressed by LepR$^+$ bone marrow cells (*Zhou et al., 2014*). Nearly all PDGFRα$^+$ cells were GFP$^+$ (*Figure 2G*). Consistent with an earlier report (*Sacchetti et al., 2007*), this suggests that *Angpt1* is widely expressed by mesenchymal stem/stromal cells in the bone marrow as LepR$^+$ cells exhibit most of the CFU-F and osteogenic activity in adult mouse bone marrow (*Zhou et al., 2014*). We were unable to detect GFP expression by endothelial cells (*Figure 2H*), or by Osteopontin$^+$ bone lining cells in the diaphysis (*Figure 2D*) or metaphysis (*Figure 2E*). Quantitative RT-PCR (qRT-PCR) analysis found that the highest levels of *Angpt1* in the bone marrow were in HSCs (350-fold higher than whole bone marrow cells [WBM]), followed by LepR$^+$ stromal cells (200-fold higher than WBM), LSK primitive HPCs (120-fold), c-kit$^+$ HPCs (60-fold), and megakaryocytes (70-fold; *Figure 2I*).

Consistent with prior studies (*Iwama et al., 1993*; *Schnurch and Risau, 1993*; *Arai et al., 2004*; *Kopp et al., 2005*), qRT-PCR showed that the Angpt1 receptor, *Tie2*, was expressed most prominently by endothelial cells (167-fold higher than WBM) and HSCs (21-fold higher; *Figure 2J*). Tie2 protein was expressed by most c-kit$^+$ HPCs and endothelial cells in normal adult bone marrow and after irradiation (*Figure 6—figure supplement 2*).

### *Angpt1* is not required for HSC maintenance

To study Angpt1 function under physiological conditions in adult bone marrow, we generated a floxed allele of *Angpt1* (*Angpt1$^{fl}$*) (*Figure 3—figure supplement 1A–C*) and an *Angpt1* deficient

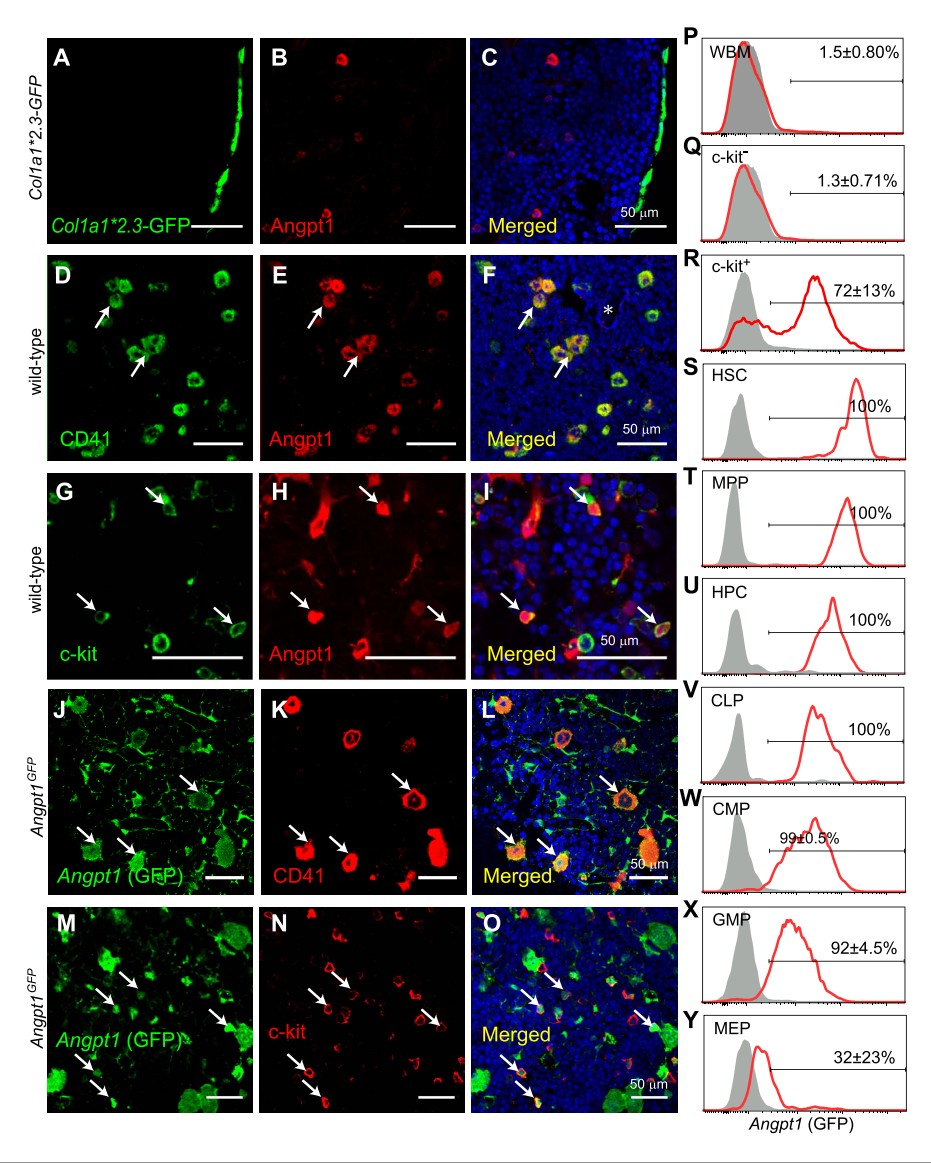

**Figure 1**. Angpt1 was expressed by megakaryocytes and hematopoietic stem/progenitor cells in the bone marrow. (**A–C**) Immunostaining of femur sections from *Col1a1\*2.3-GFP* mice with anti-Angpt1 antibody showed that Angpt1 was not detectably expressed by bone lining *Col1a1\*2.3*-GFP+ osteoblasts. Nuclei were stained with DAPI (blue). (n = 3 mice from 3 independent experiments). (**D–I**) Representative femur sections from wild-type mice showed that anti-Angpt1 antibody stained CD41+ megakaryocytes (arrows, **D–F**) and c-kit+ hematopoietic progenitors (HPCs) (arrows, **G–I**) throughout the bone marrow. * in **F** indicates trabecular bone—note the lack of Angpt1 staining in bone-lining cells (n = 3 mice from 3 independent experiments). (**J–O**) Images of femur sections from *Angpt1^GFP* mice showed that GFP was expressed by CD41+ megakaryocytes (arrows, **J–L**) and c-kit+ HPCs (arrows, **M–O**) (n = 3 mice from 3 independent experiments). (**P–Y**) Flow cytometric analysis of non-enzymatically dissociated *Angpt1^GFP* bone marrow cells (which contains hematopoietic but few stromal cells) showed that GFP was rarely expressed by whole bone marrow (WBM) cells (**P**) or c-kit− cells (**Q**) but was expressed by most c-kit+ cells (**R**), CD150+CD48−LSK hematopoietic stem cells (HSCs) (**S**), CD150−CD48−LSK multipotent progenitor cells (MPPs) (**T**), CD48+LSK HPC cells (**U**), Flt3+IL7Rα+Lineage−Sca1^low c-kit^low common lymphoid progenitors (CLPs) (**V**), CD34+FcγR−Lineage−Sca1−c-kit+ common myeloid progenitor cells (CMPs) (**W**) and CD34+FcγR+Lineage−Sca1−c-kit+ granulocyte/macrophage progenitors (GMPs) (**X**). CD34−FcγR−Lineage−Sca1−c-kit+ megakaryocytic/erythroid progenitors (MEPs) expressed little GFP (**Y**). Data represent mean ± s.d. from 4 mice from 4 independent experiments.

The following figure supplement is available for figure 1:

**Figure supplement 1**. Generation of *Angpt1^GFP* knock-in mice.

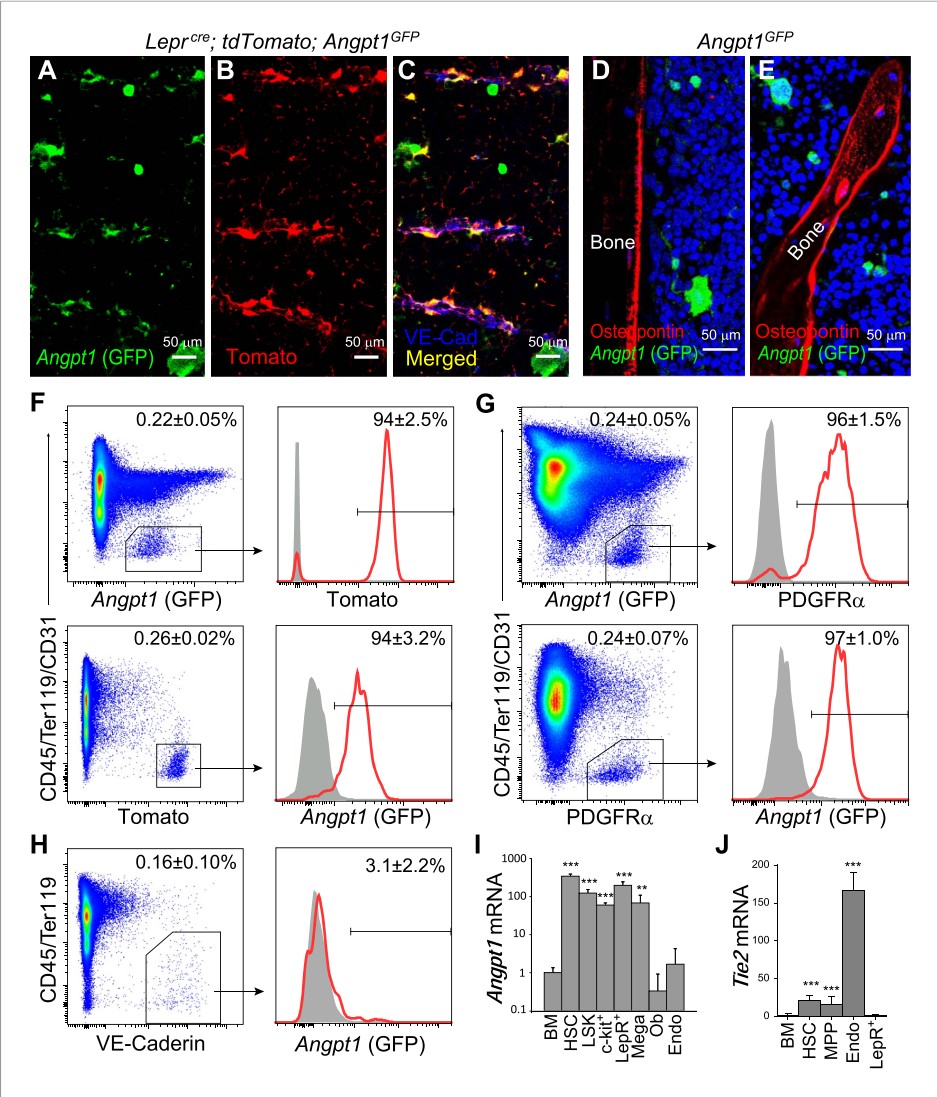

**Figure 2**. Angpt1 was expressed by Leptin Receptor[+] (LepR[+]) perivascular stromal cells but not endothelial cells or osteoblasts in bone marrow. (**A**–**C**) Representative femur sections showed that LepR[+] perivascular stromal cells (Tomato[+]) expressed GFP in *Lepr^cre*; *tdTomato*; *Angpt1^GFP* mice. Endothelial cells were stained with anti-VE-cadherin antibody (blue) (n = 3 mice from 3 independent experiments). Note that Angpt1 expression in LepR[+] cells is much easier to see in sections from GFP mice than in antibody stained sections. (**D** and **E**) Representative femur sections from *Angpt1^GFP* mice showed that GFP was not detectably expressed by Osteopontin[+] (red) osteoblasts in the diaphyseal (**D**) or metaphyseal (**E**) regions (n = 3 mice from 3 independent experiments). (**F**) In the bone marrow stroma from *Lepr^cre*; *tdTomato*; *Angpt1^GFP* mice, nearly all LepR[+] cells were positive for GFP, and vice versa. Data represent mean ± s.d. from 4 mice from 3 independent experiments. (**G**) In the bone marrow stroma from *Angpt1^GFP* mice, nearly all PDGFRα[+] cells were positive for GFP, and vice versa. Data represent mean ± s.d. from 4 mice in 3 experiments. (**H**) Bone marrow CD45[−]Ter119[−]VE-cadherin[+] endothelial cells did not express detectable GFP (n = 3 mice from 3 independent experiments). (**I** and **J**) Angpt1 (**I**) and Tie2 (**J**) transcript expression levels by qRT-PCR of unfractionated bone marrow cells, HSCs, LSK cells, c-kit[+] cells, EYFP[+] cells from *Lepr^cre*; *loxp-EYFP* mice, CD41[+] megakaryocytes, *Col1a1*2.3*-GFP[+] osteoblasts, VE-cadherin[+] bone marrow endothelial cells. All data represent mean ± s.d. from 3–8 mice/genotype in 3 independent experiments. Two-tailed Student's *t*-tests were used to assess statistical significance relative to unfractionated bone marrow cells (*p < 0.05, **p < 0.01, ***p < 0.001).

allele (*Angpt1^−*) by recombining *Angpt1^fl* in the germline using *CMV-cre*. Consistent with the embryonic lethal phenotype of two independent *Angpt1* null alleles that were previously described (*Suri et al., 1996*; *Jeansson et al., 2011*), the mating of *Angpt1^+/−* heterozygous mice did not lead to the birth of any *Angpt1^−/−* pups (*Figure 3—figure supplement 1D*). *Angpt1^−/−* embryos were found

dead when timed pregnancies were examined at E12.5 (data not shown). *Angpt1* transcripts could not be detected in the fetal livers of *Angpt1*$^{-/-}$ mice (*Figure 3—figure supplement 1E*). Thus, germline recombination of the *Angpt1*$^{fl}$ allele leads to a severe loss of *Angpt1* function.

We conditionally deleted *Angpt1* from osteoblasts using *Col1a1\*2.3-cre*; *Angpt1*$^{fl/fl}$ mice. *Col1a1\*2.3-cre* recombines efficiently in fetal and postnatal osteoblasts (*Liu et al., 2004*; *Ding et al., 2012*). *Col1a1\*2.3-cre* deleted 94 ± 3.0% of *Angpt1*$^{fl}$ alleles in *Col1a1\*2.3*-GFP$^{+}$ osteoblasts from *Col1a1\*2.3-cre*; *Angpt1*$^{fl/fl}$; *Col1a1\*2.3-GFP* mice (*Figure 3—figure supplement 2A*). Ten to 13 week-old adult *Col1a1\*2.3-cre*; *Angpt1*$^{fl/fl}$ mice had normal blood cell counts (*Figure 3—figure supplement 2B*), normal lineage composition in bone marrow, spleen and thymus (*Figure 3—figure supplement 2C*), and normal cellularity in the bone marrow, spleen and thymus (*Figure 3A* and *Figure 3—figure supplement 2D*). CD150$^{+}$CD48$^{-}$LSK HSC frequency was normal in the bone marrow and spleens of *Col1a1\*2.3-cre*; *Angpt1*$^{fl/fl}$ mice relative to littermate controls (*Figure 3B*). *Col1a1\*2.3-cre*; *Angpt1*$^{fl/fl}$ bone marrow cells also had normal frequencies of CLPs (*Figure 3—figure supplement 2E*), colony-forming progenitors in culture (*Figure 3C*) and dividing HSCs that incorporated BrdU (5′-bromo-2′-deoxyuridine) over 10 days in vivo (*Figure 3D*). *Col1a1\*2.3-cre*; *Angpt1*$^{fl/fl}$ and control bone marrow cells gave rise to similar levels of long-term multilineage reconstitution upon transplantation into irradiated mice (*Figure 3E*).

To further test whether osteolineage progenitors are a source of Angpt1 for HSC maintenance, we deleted *Angpt1* using *Osx*-Cre (*Sp7*-Cre) (*Rodda and McMahon, 2006*). *Osx*-Cre recombined 93% of *Angpt1*$^{fl}$ alleles from CD105$^{+}$PDGFRα$^{+}$CD45$^{-}$Ter119$^{-}$CD31$^{-}$ osteoprogenitors (*Park et al., 2012*) in *Osx-cre*; *Angpt1*$^{fl/fl}$ mice (*Figure 4A*). *Osx-cre*; *Angpt1*$^{fl/fl}$ mice also had normal blood cell counts (*Figure 4B*), normal cellularity, and HSC frequency in the bone marrow and spleen (*Figure 4C,D*), normal frequencies of colony-forming progenitors in culture (*Figure 4E*) and normal levels of long-term multilineage reconstitution upon transplanting bone marrow cells into irradiated mice (*Figure 4F*). Angpt1 from osteoblasts and their restricted progenitors are thus not required for hematopoiesis or HSC maintenance in normal adult mice.

To test whether Angpt1 from LepR$^{+}$ stromal cells is required for HSC maintenance we generated *Lepr*$^{cre}$; *Angpt1*$^{fl/fl}$ or *Lepr*$^{cre}$; *Angpt1*$^{fl/GFP}$ mice. *Lepr*-Cre deleted 91% of *Angpt1*$^{fl}$ alleles in LepR$^{+}$ bone marrow stromal cells from *Lepr*$^{cre}$; *Angpt1*$^{fl/GFP}$ mice (*Figure 3—figure supplement 2F*). At 8 to 13 weeks of age, *Lepr*$^{cre}$; *Angpt1*$^{fl/GFP}$ mice had normal blood cell counts (*Figure 3—figure supplement 2G*), normal lineage composition in bone marrow and spleen (data not shown), and normal cellularity in the bone marrow and spleen (*Figure 3F*). They also had normal frequencies of CD150$^{+}$CD48$^{-}$LSK HSCs in the bone marrow and spleen (*Figure 3G*), colony-forming progenitors in bone marrow (*Figure 3H*), and dividing HSCs (*Figure 3I*). *Lepr*$^{cre}$; *Angpt1*$^{fl/GFP}$ and control bone marrow cells gave rise to similar levels of long-term multilineage reconstitution upon transplantation into irradiated mice (*Figure 3J*). Angpt1 from LepR$^{+}$ stromal cells is thus not required for hematopoiesis, HSC maintenance, or HSC quiescence in normal adult mice. Similar results were obtained from adult *Nestin-cre*; *Angpt1*$^{fl/fl}$ mice (*Figure 4G–K*).

To test if Angpt1 expressed by hematopoietic cells regulates HSC function we generated *Mx1-cre*; *Angpt1*$^{fl/fl}$ mice. pIpC (polyinosinic-polycytidylic acid) was administered to mice at 2 months of age then the mice were examined 3 months later. We observed complete recombination in 98% of colonies formed by HSCs in culture (*Figure 3—figure supplement 2H*). *Mx1-cre*; *Angpt1*$^{fl/fl}$ mice had normal blood cell counts (*Figure 3—figure supplement 2I*), normal lineage composition in bone marrow and spleen (data not shown), and normal bone marrow and spleen cellularity (*Figure 3K*). *Mx1-cre*; *Angpt1*$^{fl/fl}$ mice also had normal frequencies of CD150$^{+}$CD48$^{-}$LSK HSCs in the bone marrow and spleen (*Figure 3L*), colony-forming progenitors in bone marrow (*Figure 3M*), and dividing HSCs that incorporated BrdU during a 10-day pulse (*Figure 3N*). *Mx1-cre*; *Angpt1*$^{fl/fl}$ and control bone marrow cells gave rise to similar levels of long-term multilineage reconstitution upon transplantation into irradiated mice (*Figure 3O*). Similar results were obtained when we conditionally deleted *Angpt1* from fetal hematopoietic and endothelial cells by generating *Tie2-cre*; *Angpt1*$^{fl/fl}$ mice (*Figure 5A–G*) and when we conditionally deleted *Angpt1* from megakaryocytes by generating *Pf4-cre*; *Angpt1*$^{fl/fl}$ mice (*Figure 5H–K*). Angpt1 from endothelial cells and hematopoietic cells, including megakaryocytes, is thus not required for hematopoiesis, HSC maintenance, or HSC quiescence in normal adult mice.

To globally delete *Angpt1* we generated *UBC-cre/ER*; *Angpt1*$^{fl/fl}$ mice. *UBC*-Cre/ER ubiquitously recombines in adult mice upon tamoxifen administration (*Ruzankina et al., 2007*). We administrated tamoxifen-containing chow to 8-week old *UBC-cre/ER*; *Angpt1*$^{fl/fl}$ mice as well as littermate controls for 2–5 months then sacrificed them for analysis. *UBC-cre/ER* recombined 95% of *Angpt1*$^{fl}$ alleles in

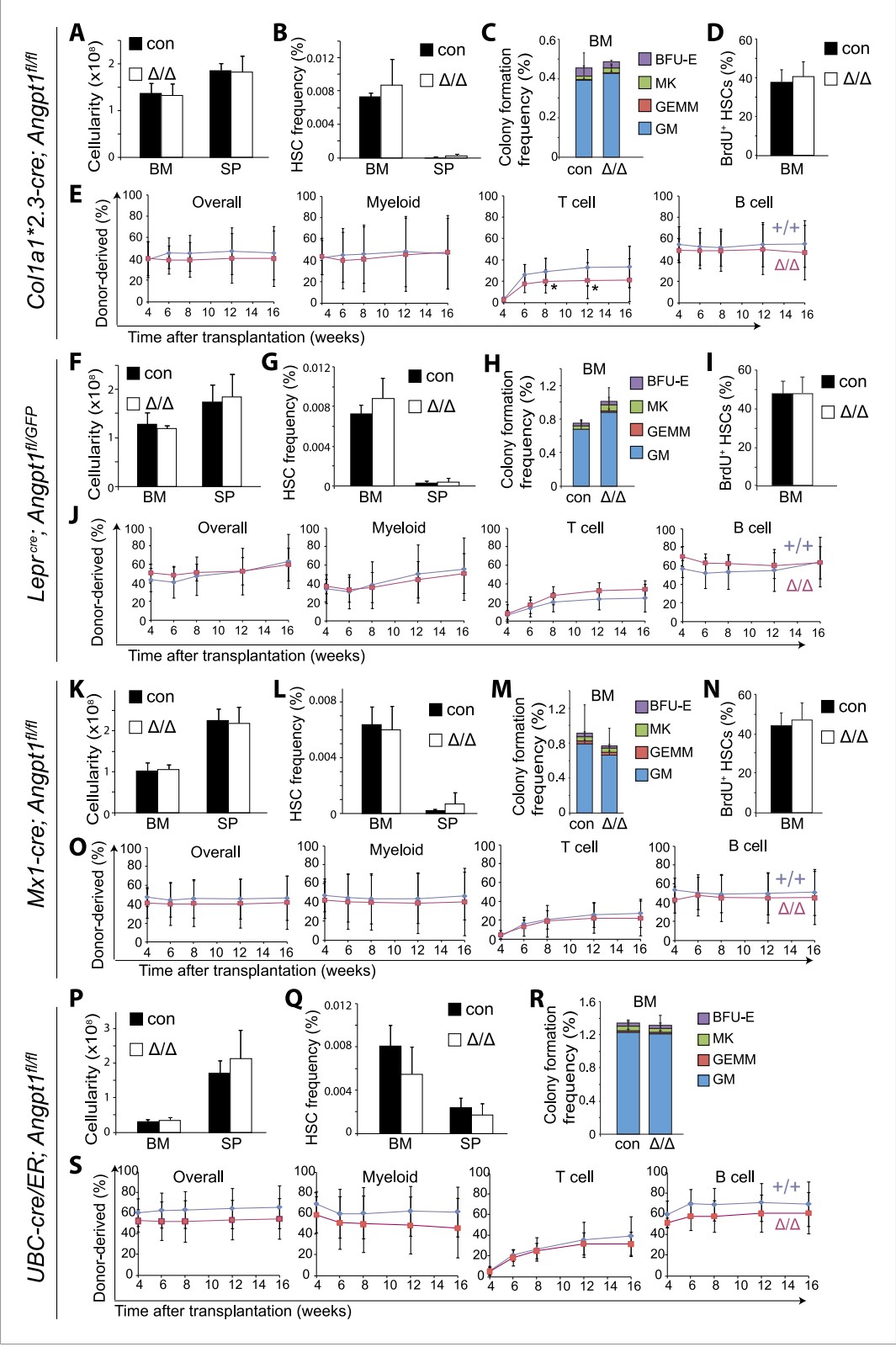

**Figure 3**. *Angpt1* was dispensable for HSC maintenance and hematopoiesis. (**A–E**) Deletion of *Angpt1* from osteoblasts using *Col1a1*2.3-cre* did not significantly affect bone marrow or spleen cellularity (**A**, n = 3 mice/genotype from 3 independent experiments), HSC frequency (**B**, n = 3 mice/genotype from 3 independent experiments), colony-forming progenitor frequency in bone marrow (**C**, n = 3 mice/genotype 3 independent
*Figure 3. continued on next page*

*Figure 3. Continued*

experiments), incorporation of a 10-day pulse of BrdU by HSCs (**D**, n = 3 pairs of male mice and 3 pairs of female mice/genotype), or reconstituting capacity of bone marrow cells in a competitive reconstitution assay (**E**, n = 14–15 recipient mice/genotype from 3 independent experiments). (**F–J**) *Lepr^cre*; *Angpt1^GFP/fl* mice had normal bone marrow and spleen cellularity (**F**, n = 4 mice/genotype from 4 independent experiments), HSC frequency in bone marrow and spleen (**G**, n = 5–6 mice/genotype from 5 independent experiments), colony-forming cell frequency in bone marrow (**H**, n = 3 mice/genotype from 3 independent experiments), BrdU incorporation into HSCs (**I**, n = 3 pairs of male mice and 3 pairs of female mice/genotype), and reconstituting capacity upon transplantation into irradiated mice (**J**, n = 13 recipient mice/genotype from 3 independent experiments). *Angpt1^fl/fl* and *Angpt1^GFP/fl* mice (lacking Cre) were indistinguishable and were therefore pooled together as controls. (**K–O**) *Mx1-cre*; *Angpt1^fl/fl* mice had normal bone marrow and spleen cellularity (**A**, n = 3 mice/genotype), HSC frequency in bone marrow and spleen (**K**, n = 3 mice/genotype), colony-forming cell frequency in bone marrow (**L**, n = 6 mice/genotype from 4 independent experiments), BrdU incorporation into HSCs (**M**, n = 3 pairs of male mice and 3 pairs of female mice/genotype), and reconstituting capacity upon transplantation into irradiated mice (**N**, n = 10–14 recipient mice/genotype from 3 independent experiments). (**P–S**) Global deletion of *Angpt1* in adult mice using *UBC*-Cre/ER (2–5 months after tamoxifen treatment) did not significantly affect cellularity in the bone marrow or spleen (**P**, n = 9–11 mice/genotype from 7 independent experiments), HSC frequency in the bone marrow (**Q**, n = 9–11 mice/genotype from 7 independent experiments), colony-forming progenitor frequency in bone marrow (**R**, n = 5 mice/genotype from 3 independent experiments), or reconstituting capacity of bone marrow cells upon transplantation into irradiated mice (**S**, n = 13–14 recipient mice/genotype from 3 independent experiments). Two-tailed Student's *t*-tests were used to assess statistical significance. See *Figure 3—figure supplement 2* for data on recombination efficiency.

The following figure supplements are available for figure 3:

**Figure supplement 1**. Generation of *Angpt1^fl* mice.

**Figure supplement 2**. Deletion of *Angpt1* did not significantly affect blood cell counts.

---

LSK cells and 96% of *Angpt1^fl* alleles in LepR^+ cells in the bone marrow of *UBC-cre/ER*; *Angpt1^fl/fl* mice (*Figure 3—figure supplement 2K*). *UBC-cre/ER*; *Angpt1^fl/fl* mice had normal blood cell counts (*Figure 3—figure supplement 2L*), normal bone marrow and spleen hematopoietic lineage composition (data not shown) and normal bone marrow and spleen cellularity relative to controls (*Figure 3P*). *UBC-cre/ER*; *Angpt1^fl/fl* mice also had normal frequencies of CD150^+CD48^−LSK HSCs in the bone marrow and spleen (*Figure 3Q*), colony-forming progenitors in bone marrow (*Figure 3R*), and long-term multilineage reconstituting bone marrow cells upon transplantation into irradiated mice (*Figure 3S*). *Angpt1* is thus dispensable for hematopoiesis and for the maintenance and function of HSCs in normal adult mice.

## *Angpt1* from LepR^+ cells and hematopoietic stem/progenitor cells delays HSC regeneration after irradiation

We next tested whether Angpt1 regulates the recovery of hematopoiesis after irradiation. Since hematopoietic cells (HSCs, c-kit^+ HPCs, and megakaryocytes) and LepR^+ stromal cells were the major sources of Angpt1 in the bone marrow (*Figures 1, 2*) we reconstituted irradiated *Lepr^cre*; *Angpt1^fl/GFP* or control recipients by transplanting $1 \times 10^6$ mechanically dissociated bone marrow cells from *Mx1-cre*; *Angpt1^fl/fl* or control donors 1 month after pIpC treatment. This allowed us to test whether Angpt1 from LepR^+ stromal cells and/or hematopoietic cells influenced the regeneration of hematopoiesis after irradiation.

We did not detect significant changes in the patterns of *Angpt1*, Tie2, or *Angpt2* expression in the bone marrow after irradiation and bone marrow transplantation (*Figure 6—figure supplement 2*). In both normal adult bone marrow, and after irradiation and transplantation, Tie2 was expressed primarily by endothelial cells and c-kit^+ HPCs (*Figure 6—figure supplement 2G*) while *Angpt2* was expressed primarily by endothelial cells.

As expected, non-irradiated adult *Lepr^cre*; *Angpt1^fl/GFP* mice, *Mx1-cre*; *Angpt1^fl/fl* mice, and *Lepr^cre*; *Mx1-cre*; *Angpt1^fl/GFP* mice all had normal bone marrow cellularity (*Figure 6A*), normal numbers of LSK cells in the bone marrow (*Figure 6B*), and normal CD150^+CD48^−LSK HSC frequency (*Figure 6C*). However, at 8 and 12 days after irradiation, *Lepr^cre*; *Angpt1^fl/GFP* mice that had been transplanted with

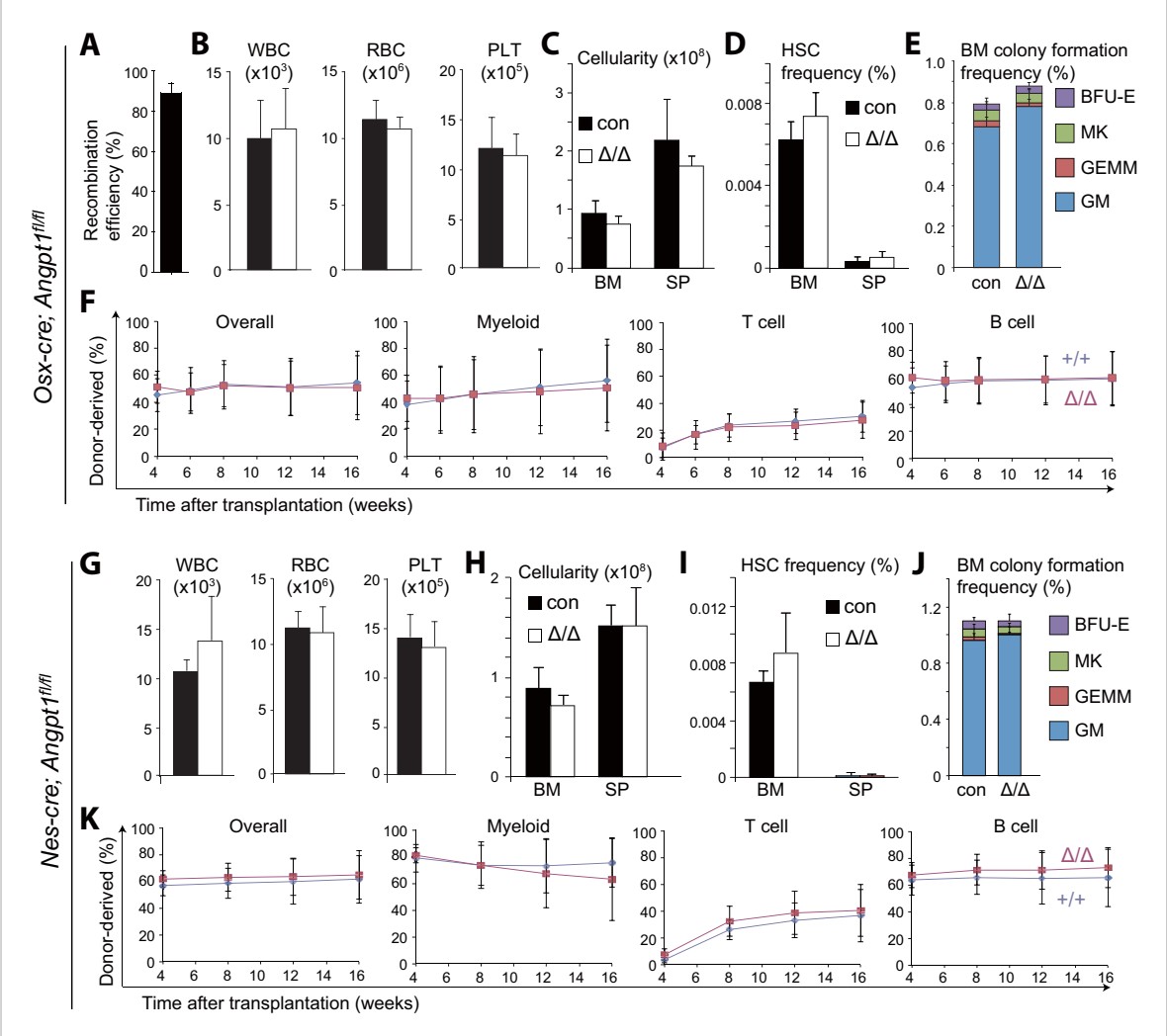

**Figure 4**. *Angpt1* from osteoblast progenitors or *Nestin*-Cre-expressing cells is dispensable for HSC maintenance and hematopoiesis. (**A**) *Osx*-Cre recombined 93 ± 3.0% of *Angpt1^fl^* alleles in CD105+PDGFRα+CD45−Ter119−CD31− osteoprogenitors from *Osx-cre; Angpt1^fl/fl^* mice. Recombination efficiency was measured as described in *Figure 3—figure supplement 2A* (n = 3 mice from 3 independent experiments). (**B–F**) *Osx-cre; Angpt1^fl/fl^* mice had normal blood cell counts (**B**, n = 6 mice/genotype from 3 independent experiments), bone marrow and spleen cellularity (**C**, n = 6–7 mice/genotype from 6 independent experiments), HSC frequency in bone marrow and spleen (**D**, n = 4 mice/genotype from 4 independent experiments), colony-forming cell frequency in bone marrow (**E**, n = 5 mice/genotype from 5 independent experiments), and reconstituting capacity upon transplantation into irradiated mice (**F**, n = 23–24 recipient mice/genotype from 5 independent experiments). (**G–K**) Young adult *Nestin-cre; Angpt1^fl/fl^* mice had normal white blood cell counts, red blood cell counts, and platelet counts (**G**, n = 4 mice/genotype from 3 independent experiments), bone marrow and spleen cellularity (**G**, n = 4 mice/genotype from 3 independent experiments), HSC frequency (**I**, n = 4 mice/genotype from 3 independent experiments), colony-forming progenitor frequency in the bone marrow (**J**, n = 3 mice/genotype), and competitive reconstituting capacity upon transplantation into irradiated mice (**K**, n = 9–10 recipient mice/genotype from 2 independent experiments). Two-tailed Student's *t*-tests were used to assess statistical significance.

*Mx1-cre; Angpt1^fl/fl^* bone marrow cells, and to a significantly lesser extent *Lepr^cre^; Angpt1^fl/GFP^* mice that had been transplanted with wild-type bone marrow cells, exhibited significantly higher bone marrow cellularity (*Figure 6A*) and significantly higher numbers of LSK cells in the bone marrow (*Figure 6B*) as compared to wild-type mice transplanted with wild-type bone marrow cells. At 16 days after irradiation, most of these differences persisted but by 28 days after irradiation mice in all treatments had similar bone marrow cellularities and LSK (Lineage−Sca1+c-kit+ cells) numbers (*Figure 6A,B*). The accelerated recovery of HPCs and hematopoiesis in the absence of *Angpt1* was also evident in white blood cell counts and in the numbers of myeloid, lymphoid, and erythroid cells in

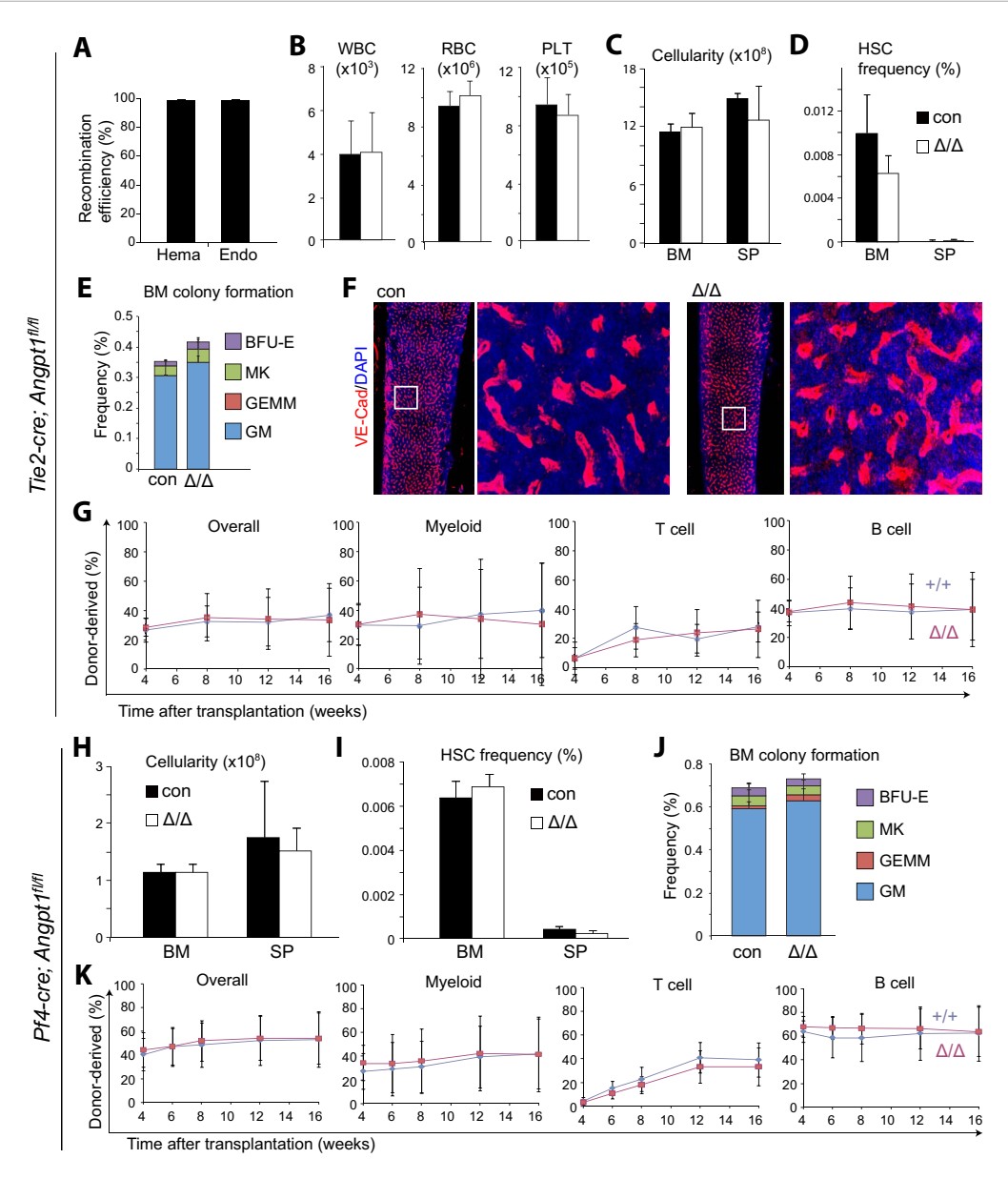

**Figure 5**. *Angpt1* from endothelial cells or megakaryocytes is dispensable for HSC maintenance and hematopoiesis. (**A**) *Tie2*-Cre recombined 97 ± 0.4% of *Angpt1*[fl] alleles in CD45[+]/Ter119[+] hematopoietic cells and 97 ± 0.6% in VE-Cadherin[+] endothelial cells from *Tie2-cre; Angpt1*[fl/fl] mice (measured as described in **Figure 3—figure supplement 2A**; n = 3 mice from 3 independent experiments). (**B–G**) *Tie2-cre; Angpt1*[fl/fl] mice had normal blood counts (**B**, n = 3–6 from 3 independent experiments), bone marrow and spleen cellularity (**C**, n = 5–10 mice/genotype from 4 independent experiments), HSC frequency in bone marrow and spleen (**D**, n = 5–10 mice/genotype from 4 independent experiments), colony-forming cell frequency in bone marrow (**E**, n = 3 mice/genotype from 3 independent experiments), vascular density and morphology (**F**, n = 3 mice/genotype from 3 independent experiments) and reconstituting capacity upon transplantation into irradiated mice (**F**, n = 8 recipient mice/genotype from 2 independent experiments). All data represent mean ± s.d. Two-tailed Student's *t*-tests were used to assess statistical significance. (**H–K**) *Pf4-cre; Angpt1*[fl/fl] mice had normal bone marrow and spleen cellularity (**H**, n = 5 mice/genotype from 4 independent experiments), HSC frequency in bone marrow and spleen (**I**, n = 5 mice/genotype from 4 independent experiments), colony-forming cell frequency in bone marrow (**J**, n = 5 mice/genotype from 5 independent experiments) and reconstituting capacity upon transplantation into irradiated mice (**K**, n = 14–15 recipient mice/genotype from 3 independent experiments).

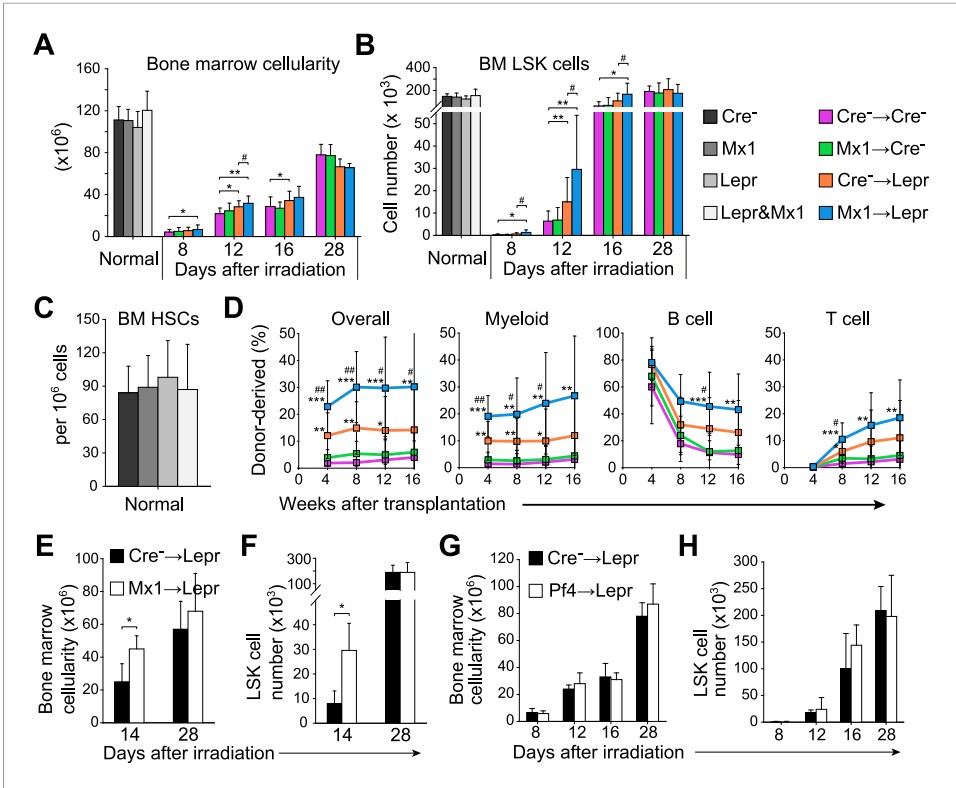

**Figure 6**. *Angpt1* deficiency in hematopoietic stem/progenitor cells and LepR⁺ stromal cells accelerated the recovery of HSCs and hematopoiesis after irradiation. One million bone marrow cells from *Angpt1^fl/fl* (Cre⁻) or *Mx1-cre; Angpt1^fl/fl* (Mx1) mice were transplanted into irradiated *Angpt1^fl/GFP* or *Angpt1^GFP* (Cre⁻) or *Lepr^cre; Angpt1^fl/GFP* (Lepr) mice (all panels reflect mean ± s.d. from 6–11 mice/genotype/time point from 5 independent experiments). Bone marrow cellularity (**A**) and LSK cell numbers (**B**) were analyzed at the indicated time points after irradiation and transplantation, always in two femurs and two tibias per mouse. (**C**) *Mx1-cre; Lepr^cre; Angpt1^fl/GFP* mice had a normal frequency of CD150⁺CD48⁻LSK HSCs in the bone marrow as compared to control (*Angpt1^fl/fl* or *Angpt1^fl/GFP*), *Mx1-cre; Angpt1^fl/fl*, or *Lepr^cre; Angpt1^fl/GFP* mice. (n = 6 mice/genotype from 5 independent experiments). (**D**) Competitive long-term multilineage reconstitution assay in which 1.5 × 10⁶ donor bone marrow cells from the indicated primary recipient mice 12 days after irradiation were transplanted along with 3 × 10⁵ recipient bone marrow cells into irradiated secondary recipient mice. The recipient cells were previously-transplanted compromised bone marrow cells. (n = 11–15 recipient mice/genotype from 3 independent experiments) Two-tailed Student's *t*-tests were used to assess statistical significance (* or #p < 0.05, ** or ##p < 0.01, *** or ###p < 0.001). * indicates statistical significance relative to Cre⁻ control cells. # indicates statistical significance relative to Mx1 cells. (**E** and **F**) 4000 LSK cells from *Angpt1^fl/fl* (Cre⁻) or *Mx1-cre; Angpt1^fl/fl* (Mx1) mice were transplanted into irradiated *Lepr^cre; Angpt1^fl/GFP* (Lepr) mice. Bone marrow cellularity (**E**) and LSK cell number in the bone marrow (**F**) were analyzed 14 and 28 days after irradiation and bone marrow transplantation. Data represent mean ± s.d. from 4 mice/genotype/time point from 3 independent experiments. Two-tailed Student's *t*-tests were used to assess statistical significance (*p < 0.05). (**G** and **H**) One million bone marrow cells from *Angpt1^fl/fl* (Cre⁻) or *Pf4-cre; Angpt1^fl/fl* (Pf4) mice were transplanted into irradiated *Lepr^cre; Angpt1^fl/GFP* (Lepr) mice. Bone marrow cellularity (**G**) and LSK cell number in the bone marrow (**H**) were analyzed at 8, 12, 16, and 28 days after irradiation and transplantation. Data represent mean ± s.d. from 4 mice/genotype/time point from 3 independent experiments. Two-tailed Student's *t*-tests were used to assess statistical significance.

The following figure supplements are available for figure 6:

**Figure supplement 1**. *Angpt1* deletion accelerated hematopoietic recovery after irradiation.

**Figure supplement 2**. *Angpt1*, Tie2, and *Angpt2* expression patterns were similar in adult bone marrow before and after irradiation.

the bone marrow (*Figure 6—figure supplement 1*). Angpt1 expression by hematopoietic cells and LepR⁺ stromal cells therefore negatively regulates the recovery of hematopoiesis after irradiation.

We tested whether Angpt1 also influenced the expansion of HSC numbers after irradiation by transplanting WBMs from mice in each of the treatments described above at 12 days after transplantation. Bone marrow cells from *Lepr^cre*; *Angpt1^fl/GFP* mice transplanted with *Mx1-cre*; *Angpt1^fl/fl* bone marrow gave significantly higher levels of donor cell reconstitution in all lineages as compared to bone marrow cells from wild-type mice transplanted with wild-type bone marrow (*Figure 6D*). To a lesser extent, bone marrow cells from *Lepr^cre*; *Angpt1^fl/GFP* mice that had been transplanted with wild-type bone marrow also gave significantly higher levels of donor cell reconstitution as compared to bone marrow cells from wild-type mice transplanted with wild-type bone marrow (*Figure 6D*). The expansion in HSC numbers during reconstitution is thus negatively regulated by Angpt1 expressed by hematopoietic cells and LepR⁺ stromal cells.

Since WBMs could potentially contain *Angpt1*-expressing stromal cells in addition to hematopoietic stem/progenitor cells and megakaryocytes we undertook a series of additional experiments to test whether hematopoietic stem/progenitor cells are a functionally important source of Angpt1 for hematopoietic regeneration. First, we transplanted 4000 LSK (Lineage⁻Sca-1⁺c-kit⁺) cells from control or *Mx1-cre*; *Angpt1^fl/fl* mice into *Lepr^cre*; *Angpt1^GFP/fl* mice to test the effects of HPCs uncontaminated by stromal cells on hematopoietic regeneration after irradiation. We found that the mice transplanted with *Angpt1* deficient LSK cells had significantly higher bone marrow cellularity (*Figure 6E*) and LSK cell numbers (*Figure 6F*) than mice transplanted with control LSK cells at 14 days after irradiation. These data prove that *Angpt1* expression by hematopoietic cells regulates hematopoietic recovery after irradiation.

The only hematopoietic cells other than c-kit⁺ hematopoietic stem and progenitor cells that express Angpt1 are megakaryocytes (*Figure 1D–F,J–L*). To test whether Angpt1 expression by megakaryocytes contributes to the regulation of hematopoietic recovery we conditionally deleted *Angpt1* from megakaryocyte lineage cells using *Pf4*-Cre and transplanted WBMs from control and *Pf4-cre*; *Angpt1^fl/fl* mice into *Lepr^cre*; *Angpt1^GFP/fl* recipients. We did not detect any significant differences in hematopoietic recovery between mice transplanted with control vs *Pf4-cre*; *Ang1^fl/fl* bone marrow (*Figure 6G,H*). These data indicate that Angpt1 expression by megakaryocyte lineage cells has little effect on hematopoietic recovery after irradiation. Together, our data demonstrate that Angpt1 expression by hematopoietic stem and progenitor cells and LepR⁺ stromal cells regulate hematopoietic recovery after irradiation.

### *Angpt1* from LepR⁺ cells and hematopoietic stem/progenitor cells delays vascular regeneration after irradiation

Consistent with a prior study (*Jeansson et al., 2011*), loss of Angpt1 expression in the bone marrow had no detectable effect on the bone marrow vasculature in normal young adult mice. Deletion of *Angpt1* from hematopoietic cells in *Mx1-cre*; *Angpt1^fl/fl* mice, or perivascular stromal cells in *Lepr^cre*; *Angpt1^fl/GFP* mice, or both in *Mx1-cre*; *Lepr^cre*; *Angpt1^fl/GFP* mice, did not affect the numbers of VE-cadherin⁺ endothelial cells or LepR⁺ perivascular cells in the bone marrow (*Figure 7A,B*), or the density or morphology of the vasculature in bone marrow relative to control mice (*Figure 7—figure supplement 1A*).

Irradiation induces vascular regression followed by regeneration from surviving endothelial cells (*Heissig et al., 2005*; *Kopp et al., 2005*; *Li et al., 2008*; *Hooper et al., 2009*). Consistent with this we observed dilated regressed sinusoids throughout the bone marrow 8 days after irradiation and bone marrow transplantation (compare *Figure 7C–F,J*). Few hematopoietic cells clustered around these regressed sinusoids relative to normal bone marrow (compare *Figure 7C–F*). Morphologically normal, hematopoietic cell-invested sinusoids were evident in some areas of the bone marrow 12 days after transplantation (*Figure 7G,J*), and their frequency increased 16 days after transplantation (*Figure 7H,J*). By day 28, regressed vessels were no longer observed in the bone marrow in any treatment (*Figure 7I,J*).

When *Mx1-cre*; *Angpt1^fl/fl* bone marrow cells were transplanted into wild-type recipients (Mx1 → Cre⁻), vascular recovery was indistinguishable from control mice (Cre⁻ → Cre⁻) (*Figure 7G,J* and *Figure 7—figure supplement 1*). However, when wild-type bone marrow cells were transplanted into *Lepr^cre*; *Angpt1^fl/GFP* recipients (Cre⁻ → Lepr) we observed accelerated morphological recovery of the vasculature, with significantly fewer dilated regressed vessels at 12 and 16 days after transplantation relative to control mice (*Figure 7G,J* and *Figure 7—figure supplement 1C*). The accelerated recovery was significantly more pronounced when we transplanted *Mx1-cre*; *Angpt1^fl/fl* bone marrow cells into *Lepr^cre*; *Angpt1^fl/GFP* recipients (Mx1 → Lepr). These mice exhibited significantly fewer

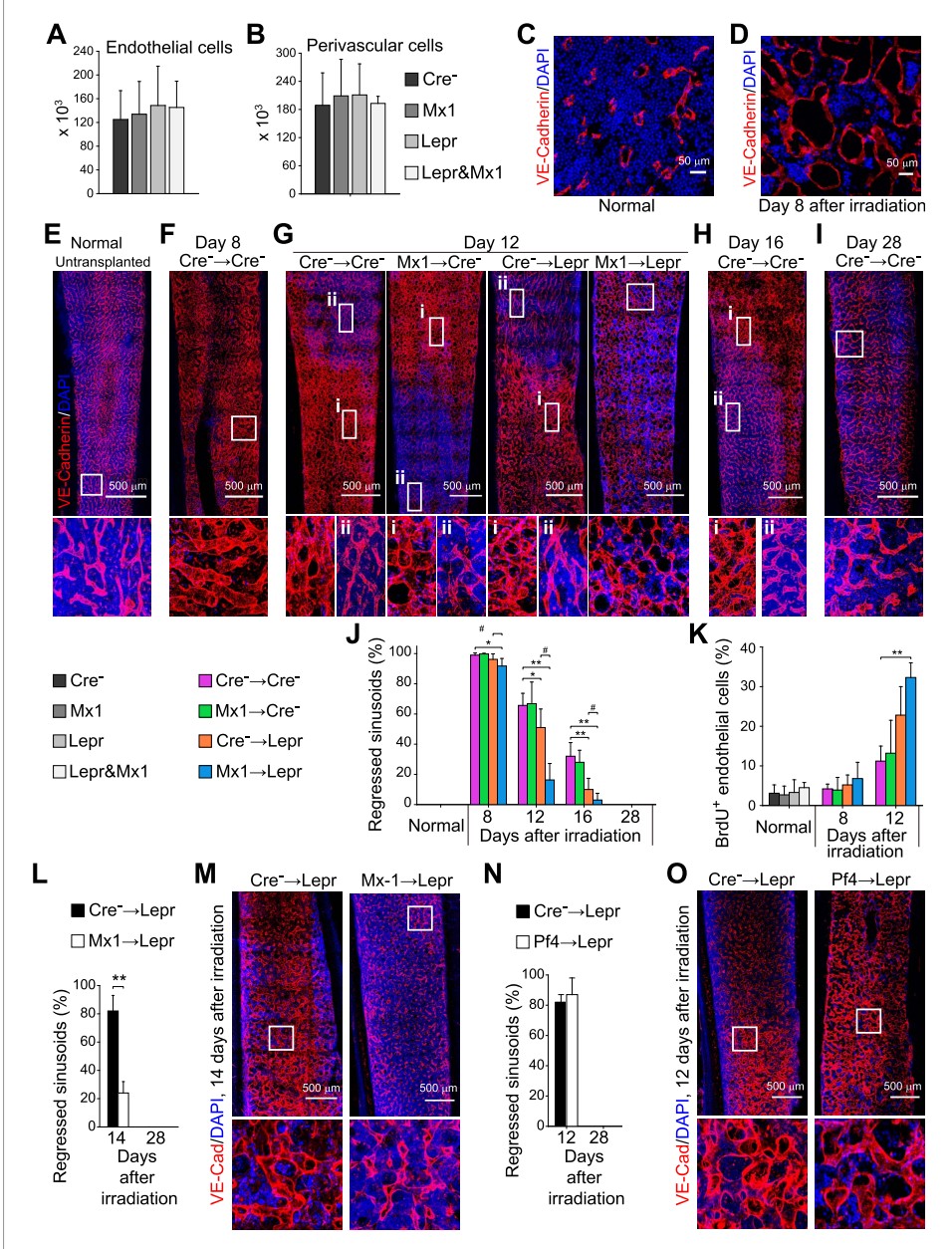

**Figure 7**. *Angpt1* deficiency in hematopoietic stem/progenitor cells and LepR[+] stromal cells increased endothelial cell proliferation and accelerated the recovery of vascular morphology after irradiation. (**A** and **B**) Deletion of *Angpt1* from hematopoietic cells (Mx1), LepR[+] stromal cells (Lepr), or both (Lepr and Mx1) did not significantly affect the number of VE-cadherin[+] endothelial cells (**A**) or LepR[+] perivascular stromal cells (**B**) in the bone marrow of normal young adult mice. Cell number in enzymatically dissociated bone marrow cells was determined in 2 pairs of femurs and tibias per mouse (n = 3 mice/genotype from 3 independent experiments). (**C** and **D**) Representative images showing normal (**C**) and regressed (**D**) sinusoids in transverse femur sections. Regressed sinusoids were distinguished from non-regressed sinusoids by being dilated and having few hematopoietic cells around them. (**E–K**) One million bone marrow cells from *Angpt1^{fl/fl}* (Cre[−]) or *Mx1-cre*; *Angpt1^{fl/fl}* (Mx1) mice were transplanted into irradiated *Angpt1^{fl/GFP}* or *Angpt1^{GFP}* (Cre[−]) or *Lepr^{cre}*; *Angpt1^{fl/GFP}* (Lepr) mice. Three-dimensional reconstructions of 50 µm thick sections of femoral bone marrow stained with anti-VE-cadherin antibody revealed the regression and regeneration of blood vessels after irradiation. Representative images for control (Cre[−]) mice were taken at steady state (**E**), 8 days (**F**), 12 days (**G**), 16 days (**H**) and 28 days (**I**) after irradiation and transplantation. Representative images for Mx1 → Cre[−], Cre[−] → Lepr and Mx1 → Lepr mice were taken 12 days after irradiation (**G**). (**J**) The percentage of regressed sinusoids in sections through the bone marrow. Data represent mean ± s.d. from 5–6

*Figure 7. continued on next page*

*Figure 7. Continued*

mice/genotype/time point from 4 independent experiments. (**K**) Incorporation of a 24-hr pulse of BrdU into VE-cadherin⁺ endothelial cells (mean ± s.d. from 3–4 mice/genotype/time from 3 experiments). Two-tailed Student's *t*-tests were used to assess statistical significance (* or #p < 0.05; ** or ##p < 0.01; *** or ###p < 0.001). (**L** and **M**) 4000 LSK cells from *Angpt1^{fl/fl}* (Cre⁻) or *Mx1-cre*; *Angpt1^{fl/fl}* (Mx1) mice were transplanted into irradiated *Lepr^{cre}*; *Angpt1^{fl/GFP}* (Lepr) mice. Vascular morphology (**M**) and the percentage of regressed sinusoids (**L**) were analyzed at the indicated time points. (**N** and **O**) One million bone marrow cells from *Angpt1^{fl/fl}* (Cre⁻) or *Pf4-cre*; *Angpt1^{fl/fl}* (Pf4) mice were transplanted into irradiated *Lepr^{cre}*; *Angpt1^{fl/GFP}* (Lepr) mice. Vascular morphology (**O**) and the percentage of regressed sinusoids (**N**) were analyzed at the indicated time points. Two-tailed Student's *t*-tests were used to assess statistical significance (*p < 0.05).

The following figure supplement is available for figure 7:

**Figure supplement 1**. *Angpt1* deficiency accelerated the recovery of vascular morphology after irradiation.

regressed vessels at 12 and 16 days after transplantation relative to control mice and Cre⁻ → Lepr mice (*Figure 7G,J* and *Figure 7—figure supplement 1C*). By 28 days after irradiation mice in all treatments had reacquired morphologically normal bone marrow vasculature (*Figure 7I,J* and *Figure 7—figure supplement 1D*). These data indicate that Angpt1 produced by hematopoietic cells and LepR⁺ stromal cells slows the morphological recovery of blood vessels after irradiation. When combined with the observation that Angpt1 also slows the regeneration of HSCs (*Figure 6B,D*) and bone marrow hematopoiesis (*Figure 6A*) after irradiation, the data indicate that Angpt1 negatively regulates the regeneration of the HSC niche in bone marrow after irradiation.

We transplanted 4000 LSK cells from control and *Mx1-cre*; *Angpt1^{fl/fl}* mice into *Lepr^{cre}*; *Angpt1^{GFP/fl}* mice to test the effects of HPCs uncontaminated by stromal cells on vascular regeneration after irradiation. We found that the mice transplanted with *Angpt1* deficient LSK cells had significantly better vascular morphology in the bone marrow (*Figure 7L,M*) than mice transplanted with control LSK cells at 14 days after irradiation. To test whether Angpt1 expression by megakaryocytes contributed to the vascular recovery we transplanted WBMs from control and *Pf4-cre*; *Angpt1^{fl/fl}* mice into *Lepr^{cre}*; *Angpt1^{GFP/fl}* recipients. We did not detect any significant differences in vascular recovery between mice transplanted with control vs *Pf4-cre*; *Ang1^{fl/fl}* bone marrow (*Figure 7N,O*). Angpt1 expression by hematopoietic stem and progenitor cells and LepR⁺ stromal cells therefore regulate both hematopoietic and vascular recovery after irradiation.

To investigate the cellular mechanism by which Angpt1 influences vascular recovery after irradiation we assessed the proliferation of bone marrow endothelial cells. In normal adult bone marrow few endothelial cells incorporated a 24-hr pulse of BrdU and deletion of *Angpt1* from hematopoietic cells, LepR⁺ cells, or both did not influence this frequency (*Figure 7K*). After irradiation and bone marrow transplantation, endothelial cells were recruited into cycle (*Figure 7K*). Deletion of *Angpt1* from hematopoietic cells and LepR⁺ cells significantly increased the frequency of dividing endothelial cells 12 days after transplantation (*Figure 7K*). These data suggest that Angpt1 slows the recovery of the vasculature and the HSC niche partly by negatively regulating the proliferation of endothelial cells after irradiation.

## *Angpt1* from LepR⁺ cells and hematopoietic stem/progenitor cells promotes vascular integrity during regeneration

To test whether Angpt1 regulates vascular leakiness in the bone marrow we assessed Evans blue extravasation. Evans blue binds to serum albumin and can be used to trace macromolecule flux across blood vessels (*Radu and Chernoff, 2013*). In normal bone marrow we observed little Evans blue extravasation, irrespective of whether *Angpt1* was deleted from hematopoietic cells, LepR⁺ cells, or both (*Figure 8A*), suggesting that *Angpt1* is dispensable for maintaining vascular integrity in normal adult bone marrow. In contrast, 12 days after irradiation we observed uniformly high levels of Evans blue extravasation in bone marrow from mice in all treatments (*Figure 8A* and *Figure 8—figure supplement 1A*), consistent with the leakiness that would be expected from regenerating blood vessels (*Hooper et al., 2009*). When morphological recovery of bone marrow vessels was complete 28 days after irradiation (*Figure 7J* and *Figure 7—figure supplement 1D*), control mice (Cre⁻ → Cre⁻) and wild-type mice transplanted with *Angpt1* deficient bone marrow cells (Mx1 → Cre⁻) had largely

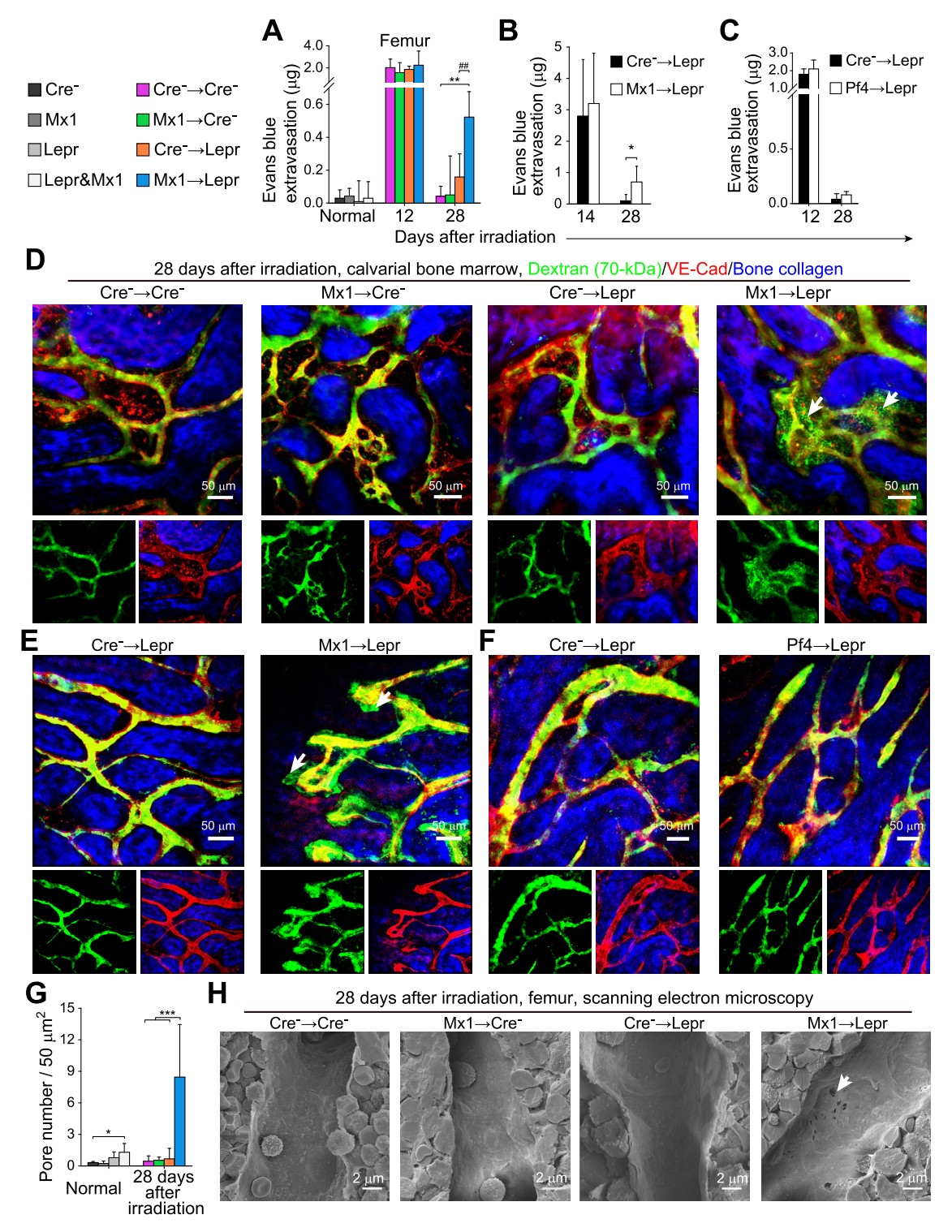

**Figure 8**. *Angpt1* deficiency in hematopoietic stem/progenitor cells and LepR$^+$ stromal cells increases the leakiness of regenerated blood vessels. One million bone marrow cells from *Angpt1$^{fl/fl}$* (Cre$^-$) or *Mx1-cre; Angpt1$^{fl/fl}$* (Mx1) mice were transplanted into lethally irradiated *Angpt1$^{GFP/fl}$* or *Angpt1$^{GFP}$* (Cre$^-$) or *Lepr$^{cre}$; Angpt1$^{GFP/fl}$* (Lepr) mice (**A**, **D**, **H**) (n = 4 mice/genotype/time point from 3 independent experiments). 4000 LSK cells *Angpt1$^{fl/fl}$* (Cre$^-$) or *Mx1-cre; Angpt1$^{fl/fl}$* (Mx1) mice were transplanted into *Lepr$^{cre}$; Angpt1$^{GFP/fl}$* mice (Lepr) (**B** and **E**) (n = 4 mice/genotype/time point from 3 independent experiments). One million bone marrow cells from *Angpt1$^{fl/fl}$* (Cre$^-$) or *Pf4-cre; Angpt1$^{fl/fl}$* (Pf4) were transplanted into *Lepr$^{cre}$; Angpt1$^{GFP/fl}$* (Lepr) mice (**C** and **F**) (n = 4 mice/genotype/time point from 3 independent experiments). (**A**–**C**) Extravasation of intravenously injected Evans blue into femoral bone

*Figure 8. continued on next page*

Figure 8. Continued

marrow at the indicated time points after irradiation and bone marrow transplantation. (**D**–**F**) Live imaging of calvarial bone marrow at 28 days after irradiation and bone marrow transplantation to assess dextran-FITC extravasation (arrows). The mice were injected with dextran-FITC and anti-VE-cadherin antibody before microscopy. (**G**) Quantification of the number of pores larger than 100 nm in diameter per 50 µm$^2$ of sinusoidal endothelium (n = 3–5 mice/genotype from 3 independent experiments). Two-tailed Student's $t$-tests were used to assess statistical significance (* or #, p < 0.05; ** or ##, p < 0.01; *** or ###, p < 0.001). (**H**) Scanning electron microscopy of bone marrow sinusoids from Cre⁻ → Cre⁻, Mx1 → Cre⁻, Cre⁻ → Lepr, and Mx1 → Lepr mice at 28 days after irradiation. Arrows indicate pores greater than 100 nm in diameter in sinusoidal endothelium 28 days after irradiation.

The following figure supplement is available for figure 8:

**Figure supplement 1**. *Angpt1* deficiency led to the persistence of pores and leakiness in blood vessels after irradiation.

re-established vascular integrity, with little Evans blue extravasation (*Figure 8A*). *Lepr^cre*; *Angpt1^fl/GFP* mice transplanted with wild-type bone marrow cells (Cre⁻ → Lepr) showed a trend toward increased Evans blue extravasation but the effect was not statistically significant relative to control mice (*Figure 8A*). In contrast, *Lepr^cre*; *Angpt1^fl/GFP* mice transplanted with *Angpt1* deficient bone marrow cells (Mx1 → Lepr) exhibited significantly higher levels of Evans blue extravasation at 28 days after transplantation (*Figure 8A*). Leaky vasculature was not observed in the spleen (*Figure 8—figure supplement 1B*). Angpt1 from hematopoietic cells and LepR⁺ stromal cells is thus required to promote vascular integrity in the bone marrow after regeneration at the expense of slowing endothelial cell proliferation and the morphological recovery of blood vessels, slowing the regeneration of the HSC niche.

To independently assess vascular integrity, we performed live imaging of the vasculature in the calvarium bone marrow of mice intravenously administered anti-VE-cadherin antibody and dextran-FITC (70 kDa). In normal mice, dextran-FITC fluorescence was tightly restricted within VE-cadherin⁺ vessels (*Figure 8—figure supplement 1C*). 12 days after irradiation, calvarium blood vessels became dilated and dextran-FITC leaked throughout the medullary cavity (*Figure 8—figure supplement 1D*). At 28 days after transplantation, control mice (Cre⁻ → Cre⁻), wild-type mice transplanted with *Angpt1* deficient hematopoietic cells (Mx1 → Cre⁻), and *Lepr^cre*; *Angpt1^fl/GFP* mice transplanted with wild-type bone marrow cells (Cre⁻ → Lepr) all exhibited vascular integrity, with little discernible leakage of dextran-FITC (*Figure 8D*). In contrast, *Lepr^cre*; *Angpt1^fl/GFP* mice transplanted with *Angpt1* deficient bone marrow cells (Mx1 → Lepr) exhibited leaky vessels in the calvarium with dextran-FITC infiltrating the bone marrow (*Figure 8D*, see arrows). Thus, consistent with the Evans blue assay, Angpt1 from hematopoietic cells and LepR⁺ cells promotes bone marrow vascular integrity during regeneration.

We transplanted 4000 LSK cells from control and *Mx1-cre*; *Angpt1^fl/fl* mice into *Lepr^cre*; *Angpt1^GFP/fl* mice to test the effects of HPCs uncontaminated by stromal cells on vascular integrity after irradiation. The mice transplanted with *Angpt1* deficient, but not control, LSK cells exhibited vascular leakage in the bone marrow 28 days after irradiation, as evidenced by high level of Evans blue and Dextran-FITC extravasation (*Figure 8B,E*). To test whether Angpt1 expression by megakaryocytes contributed to the vascular integrity we transplanted WBMs from control and *Pf4-cre*; *Angpt1^fl/fl* mice into *Lepr^cre*; *Angpt1^GFP/fl* recipients. We did not detect any significant differences in vascular leakiness between mice transplanted with control vs *Pf4-cre*; *Ang1^fl/fl* bone marrow (*Figure 8C,F*). Angpt1 expression by hematopoietic stem and progenitor cells and LepR⁺ stromal cells therefore promotes vascular integrity during regeneration after irradiation.

We performed scanning electron microscopy to better understand the loss of blood vessel integrity after *Angpt1* deletion. Normal bone marrow sinusoids were 5–20 µm in luminal diameter (*Figure 8—figure supplement 1E*). They could be readily distinguished from bone marrow arterioles, which had thicker walls and a different morphology (*Figure 8—figure supplement 1G*). At 8–16 days after irradiation, sinusoid diameter in the bone marrow increased (compare *Figure 8—figure supplement 1E,F*) and the endothelial lining was marked by small pores (*Figure 8—figure supplement 1F*), consistent with the finding that irradiation causes discontinuities in bone marrow blood vessels (*Daldrup et al., 1999*). At 28 days after irradiation, pores were rare in sinusoids from wild-type mice transplanted with wild-type marrow (Cre⁻ → Cre⁻), wild-type mice transplanted with *Angpt1* deficient bone marrow (Mx1 → Cre⁻), and *Lepr^cre*; *Angpt1^fl/GFP* mice transplanted with wild-type bone marrow (Cre⁻ → Lepr) (*Figure 8G,H*), consistent with the integrity of vessels in these mice. In contrast, pores remained common in blood vessels in bone marrow from *Lepr^cre*; *Angpt1^fl/GFP* mice

transplanted with *Angpt1* deficient bone marrow cells (Mx1 → Lepr) (*Figure 8H*, see arrow; *Figure 8G*). These data suggest that in the absence of Angpt1, vascular integrity is reduced in regenerating bone marrow blood vessels because of the persistence of pores or discontinuities among endothelial cells.

## Vascular leakiness does not accelerate hematopoietic recovery

To test whether the accelerated hematopoietic recovery in the absence of *Angpt1* is caused by the increase in vascular leakiness we treated mice with cavtratin, an anti-permeability agent unrelated to Angpt1 function (*Gratton et al., 2003*). We injected cavtratin (2.5 mg/kg/day i.p.) into control (control bone marrow transplanted into control mice) and Mx1 → Lepr recipients from 7 to 13 days after irradiation then analyzed the mice 14 days after irradiation and bone marrow transplantation. Based on both Dextran-FITC live-imaging and Evans blue extravasation, cavtratin significantly reduced vascular leakiness in both control and Mx1 → Lepr recipient mice (*Figure 9A,B*). However, cavtratin administration did not significantly affect the recovery of bone marrow cellularity or LSK cell numbers in the bone marrow of control or Mx1 → Lepr recipients (*Figure 9C,D*). Mx1 → Lepr recipients continued to regenerate bone marrow cellularity and LSK cell numbers significantly faster than control mice, irrespective of cavtratin treatment. The accelerated recovery of hematopoietic stem/progenitor cells and hematopoiesis in the absence of *Angpt1* is therefore not caused by increased vascular leakiness. These appear to reflect independent effects of *Angpt1*.

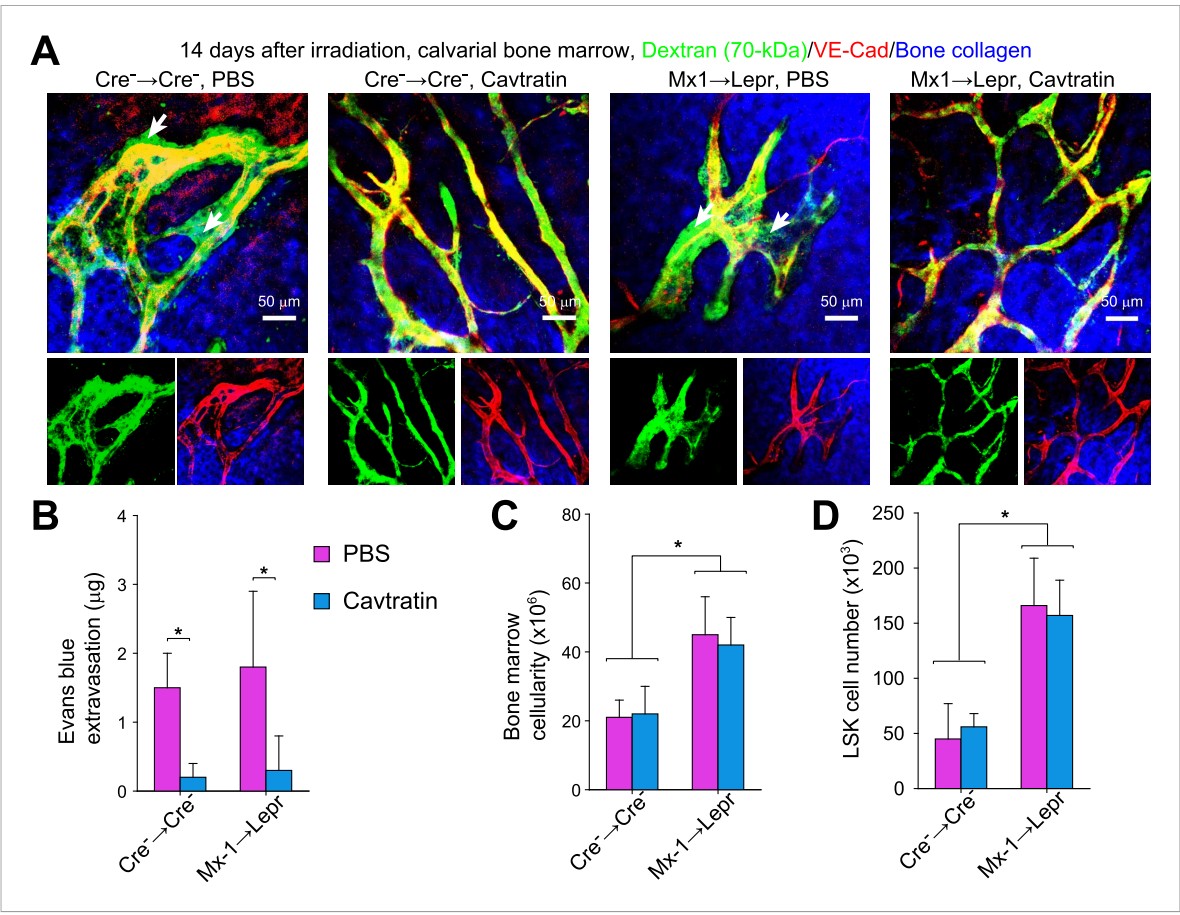

**Figure 9**. Vascular leakage does not promote hematopoietic regeneration in *Angpt1* mutant mice. One million whole WBMs from control and *Mx1-cre; Angpt1*$^{fl/fl}$ mice were transplanted into irradiated control and *Lepr*$^{cre}$; *Angpt1*$^{GFP/fl}$ mice, respectively. Cavtratin was administered into control (Cre⁻ → Cre⁻), and mutant (Mx1 → Lepr) recipients from 7 to 13 days after irradiation and transplantation. 14 days after irradiation mice were analyzed for Dextran-FITC extravasation in calvarial bone marrow (**A**, n = 3 mice/genotype from 3 independent experiments), Evans blue extravasation in femoral bone marrow (**B**, n = 3 mice/genotype from 3 independent experiments), bone marrow cellularity, and LSK cell number in the bone marrow (**C** and **D**, n = 4 mice/genotype from 3 independent experiments). Cell numbers reflect two femurs and two tibias. Two-tailed Student's *t*-tests were used to assess statistical significance (*p < 0.05).

## Discussion

Several gene-expression profiling studies are consistent with our finding that HSCs are a major source of Angpt1 (*Ivanova et al., 2002*; *Akashi et al., 2003*; *Forsberg et al., 2010*; *Cabezas-Wallscheid et al., 2014*). In addition, an *Angpt1*-LacZ knockin allele is expressed in an HSC-enriched population (*Takakura et al., 2000*). Several prior studies have also demonstrated that perivascular stromal cells are a significant source of Angpt1 in the bone marrow (*Sacchetti et al., 2007*; *Mendez-Ferrer et al., 2010*; *Ding et al., 2012*). In contrast, we have been unable to detect *Angpt1* expression in osteoblasts in *Angpt1^GFP* knock-in mice (*Figure 2D,E*), by anti-Angpt1 antibody staining (*Figure 1A–C*), by qRT-PCR on sorted cells (*Figure 2I*), or by gene-expression profiling of osteoblasts (data not shown). Megakaryocytes also express Angpt1 (*Figure 1J–L*) but unlike deletion in hematopoietic stem/progenitor cells and LepR+ stromal cells, conditional deletion in megakaryocyte lineage cells did not affect hematopoietic or vascular regeneration after irradiation (*Figures 6–9*). Thus, hematopoietic stem/progenitor cells and LepR+ stromal cells are the functionally important sources of Angpt1 in the bone marrow.

Tie1 and Tie2 are the receptors for Angpt1, Angpt2, and possibly Angpt3 (*Augustin et al., 2009*; *Eklund and Saharinen, 2013*; *Fagiani and Christofori, 2013*; *D'Amico et al., 2014*; *Thomson et al., 2014*). *Tie1* is not required for the development or maintenance of fetal liver or adult bone marrow HSCs (*Corash et al., 1989*; *Rodewald and Sato, 1996*). Tie1/Tie2 double knockout ES cells contribute to fetal but not adult hematopoiesis (*Puri and Bernstein, 2003*). Given that *Angpt1* is not required for the maintenance of adult hematopoiesis, some combination of Angpt2 and Angpt3 may be required for adult hematopoiesis.

Global *Angpt1* deletion (*Figure 3P–S*), deletion from osteoblasts and their progenitors (*Figure 3A–E* and *Figure 4A–F*), or deletion from hematopoietic and/or LepR+ stromal cells (*Figure 3F–O*), did not affect HSC frequency or HSC function in normal adult mice. *Angpt1* deletion from these cell populations had little effect on the bone marrow vasculature in normal adult mice (*Figures 7, 8*). We observed only rare pores in sinusoidal epithelium from *Lepr^cre*; *Mx1-cre*; *Angpt1^fl/GFP* mice (*Figure 8G,H*) and virtually no Evans blue leakage (*Figure 8A*). In contrast, *Angpt1* deletion from LepR+ stromal cells and hematopoietic cells had much larger effects on the regeneration of the vasculature and the HSC niche after irradiation. *Angpt1* deficiency from these cell populations accelerated the recovery of hematopoiesis (*Figure 6A*), the regeneration of LSK cells (*Figure 6B*), the regeneration of long-term multilineage reconstituting HSCs (*Figure 6D*), the proliferation of endothelial cells (*Figure 7K*), and the morphological recovery of bone marrow blood vessels (*Figure 7E–J* and *Figure 7—figure supplement 1*). However, *Angpt1* deficiency also increased the leakiness of the regenerated blood vessels (*Figure 8A,D*) by allowing pores to persist in sinusoidal endothelium (*Figure 8G,H*). Together, the data indicate that Angpt1 produced by LepR+ stromal cells and hematopoietic cells promotes vascular integrity in regenerating blood vessels in the bone marrow at the cost of slowing the regeneration of HSC niches and hematopoiesis.

## Materials and methods

### Mice

Targeting vectors for generating *Angpt1^GFP* and *Angpt1^fl/+* mice were constructed by recombineering (*Liu et al., 2003*). Linearized targeting vectors were electroporated into Bruce4 ES cells. Corrected targeted ES cell clones were identified by Southern blotting and injected into C57BL/6-Tyr^c-2J blastocysts. The resulting chimeric mice were bred with C57BL/6-Tyr^c-2J mice to obtain germline transmission. Then the *Frt-Neo-Frt* cassette introduced by the targeting vector was removed by mating with Flpe mice (*Rodriguez et al., 2000*). These mice were backcrossed onto a C57BL/Ka background. Other mice used in this study were: *Col1a1*2.3-cre* (*Liu et al., 2004*), *Lepr^cre* (*DeFalco et al., 2001*), *Osx-cre* (*Rodda and McMahon, 2006*), *Nestin-cre* (*Tronche et al., 1999*), *Pf4-cre* (*Tiedt et al., 2007*), *Tie2-cre* (*Koni et al., 2001*), *Mx1-cre* (*Kühn et al., 1995*), *UBC-cre/ER* (*Ruzankina et al., 2007*), *Col1a1*2.3-GFP* (*Kalajzic et al., 2002*), *Loxp-EYFP* (*Srinivas et al., 2001*), and *Loxp-tdTomato* (*Madisen et al., 2010*). For induction of *UBC*-Cre/ER, Tamoxifen chow (Harlan, Indianapolis, IN) containing tamoxifen citrate at 40 mg/kg, with 5% sucrose, was administrated to mice for 2–5 months before analysis. C57BL/6-SJL (CD45.1) mice were used as recipients in transplantation experiments unless otherwise indicated. All mice were housed at the Unit for Laboratory Animal Medicine at the University of Michigan or in the Animal Resource Center at the University of Texas Southwestern

Medical Center. All protocols were approved by the University of Michigan Committee on the Use and Care Animals and by the University of Texas Southwestern Institutional Animal Care and Use Committee.

## Genotyping PCR

The following primers were used for genotyping. For *Angpt1GFP*, OLD308: 5′-gggaaagagtcaaacaagcag-3′ OLD309: 5′-aaccgaaagcgatcacttac-3′ and OLD292: 5′-cggacacgctgaacttgtgg-3′. For *Angpt1fl*, OLD335: 5′-ggactcaacttcctgggtaagc-3′ and OLD336: 5′-ggctttgacagtcaaaatgcc-3′. For *Angpt1⁻*, OLD3111: 5′-cag ggttggcataaaatttgc-3′ and OLD350: 5′-tcctggtctttgcacttgactg-3′.

## Quantitative reverse transcription PCR

Cells were directly sorted into Trizol. Total RNA was extracted per the manufacture's instructions. SuperScript III (Lifetech, Grand Island, NY) was used to generate cDNA. Quantitative real-time PCR was performed using SYBR green on a LightCycler 480 or Stepone Plus. β-actin amplification was used to normalize the transcript content of samples. Primers used in this study were: *Angpt1*: OLD7: 5′-gggggaggttggacagtaat-3′ and OLD8: 5′-catcagctcaatcctcagca-3′. *Tie2*: forward: 5′-gattttggattgtc ccgaggtcaag-3′ and reverse: 5′-caccaatatctgggcaaatgatgg-3′. *β-actin*: OLD27: 5′-gctcttttccagccttcctt-3′ OLD28: 5′-cttctgcatcctgtcagcaa-3′.

## Methylcellulose culture

Cells were sorted or directly pipetted into methylcellulose culture medium (3434, Stemcell Technologies, Vancouver, BC, Canada) and incubated at 37°C for 14 days in a humidified chamber.

## Flow cytometry

Bone marrow cells were isolated by flushing or by crushing the long bones with a mortar and pestle in $Ca^{2+}$ and $Mg^{2+}$ free HBSS with 2% heat-inactivated bovine serum. Spleen cells were obtained by crushing the spleen between two glass slides. The cells were dissociated to a single cell suspension by gently passing through a 25G needle then filtering through a 70 μm nylon mesh. The following antibodies were used to isolate HSCs: anti-CD150 (TC15-12F12.2), anti-CD48 (HM48-1), anti-Sca1 (E13-161.7), anti-c-kit (2B8), and the following antibodies against lineage markers (anti-Ter119, anti-B220 [6B2], anti-Gr1 [8C5], anti-CD2 [RM2-5], anti-CD3 [17A2], anti-CD5 [53-7.3], and anti-CD8 [53-6.7]). HPCs were identified by flow cytometry using the following antibodies: anti-Sca1 (E13-161.7), anti-c-kit (2B8), and the following antibodies against lineage markers (anti-Ter119, anti-B220 [6B2], anti-Gr1 [8C5], anti-CD2 [RM2-5], anti-CD3 [17A2], anti-CD5 [53-7.3] and anti-CD8 [53-6.7]), anti-CD34 (RAM34), anti-CD135 (Flt3) (A2F10), anti-CD16/32 (FcγR) (93), anti-CD127 (IL7Rα) (A7R34), anti-CD24 (M1/69), anti-CD43 (1B11), anti-B220 (6B2), anti-IgM (II/41), anti-CD3 (17A2), anti-Gr1 (8C5), anti-Mac1 (M1/70), anti-CD41 (MWReg30), anti-CD71 (C2), anti-Ter119, anti-CD44 (IM7) and anti-CD25 (PC61). DAPI was used to exclude dead cells. Unless otherwise indicated, antibodies were obtained from eBioscience (San Diego, CA) or BD Bioscience (San Jose, CA).

For flow cytometric analysis of bone marrow stromal cells, bone marrow was flushed using HBSS with 2% bovine serum. Then, whole bone marrow was digested with DNase I (200 U/ml) and Collagenase IV (200 U/ml) or liberase (Roche, San Francisco, CA) at 37°C for 15 min. Samples were then stained with antibodies and analyzed by flow cytometry. Anti- PDGFRα (APA5), anti-CD45 (30F-11), anti-CD31 (390), and anti-Ter119 antibodies were used to isolate perivascular stromal cells. For analysis of bone marrow endothelial cells, mice were i.v. injected with 10 μg/mouse Alexa Fluor 647 conjugated anti-VE-cadherin antibody (BV13, eBiosciences) (*Butler et al., 2010*). 10 min later, the long bones were removed and bone marrow was flushed, digested, and stained as above. Samples were analyzed using a FACSAria or FACSCanto II flow cytometer (BD Biosciences). Data were analyzed by FACSDiva (BD Biosciences) or FlowJo (Tree Star) software.

## Long-term competitive reconstitution assay

Adult recipient mice were lethally irradiated by a Cesium 137 GammaCell40 Irradiator (MDS Nordia) or an XRAD 320 x-ray irradiator (Precision X-Ray Inc., North Branford, CT) with two doses of 540 rad (total 1080 rad) delivered at least 2 hr apart. Cells were transplanted intravenously into the retro-orbital venous sinus of anesthetized mice. $3 \times 10^5$ bone marrow cells were transplanted together with $3 \times 10^5$ recipient type competitor cells unless otherwise noted. Mice were maintained on antibiotic water (neomycin sulfate 1.11 g/l and polymixinB 0.121 g/l) for 14 days then switched to regular water.

Recipient mice were periodically bled to assess the level of donor-derived blood cells, including myeloid, B and T cells for at least 16 weeks. Blood was subjected to ammonium chloride/potassium red cell lysis before antibody staining. Antibodies including anti-CD45.2 (104), anti-CD45.1 (A20), anti-Gr1 (8C5), anti-Mac-1 (M1/70), anti-B220 (6B2), and anti-CD3 (KT31.1) were used to stain cells for analysis by flow cytometry.

## Cell cycle analysis

For BrdU incorporation assays, mice were given an intraperitoneal injection of 1 mg BrdU (Sigma, St. Louis, MO) per 6 g of body mass in PBS (Phosphate Buffered Saline) and maintained on 1 mg/ml of BrdU in the drinking water for 24 hr (endothelial cells) or 10 days (HSCs). Bone marrow endothelial cells were pre-stained by i.v. injection of Alexa Fluor 555 conjugated anti-VE-cadherin antibody (BV13, eBiosciences). The frequency of BrdU$^+$ cells was then analyzed by flow cytometry using the APC BrdU Flow Kit (BD Biosciences).

## Bone sectioning, immunostaining, and confocal imaging

Freshly dissected bones were fixed in 4% paraformaldehyde overnight followed by 3-day decalcification in 10% EDTA. Bones were sectioned using the CryoJane tape-transfer system (Instrumedics, St. Louis, MO). Sections were blocked in PBS with 10% horse serum for 1 hr and then stained overnight with goat-anti-Angpt1 (Santa Cruz, Dallas, TX, 1:200), chicken-anti-GFP (Aves, Tigard, OR, 1:1000), anti-CD41-PE (eBioscience, clone eBioMWReg30, 1:200) and/or goat-anti-Osteopontin (R&D, Minneapolis, MN, 1:400) antibodies. Donkey-anti-goat Alexa Fluor 647, donkey-anti-chicken Alexa Fluor 488, and/or Donkey-anti-goat Alexa Fluor 555 were used as secondary antibodies (Invitrogen, Grand Island, NY, 1:400). Slides were mounted with anti-fade prolong gold (Invitrogen) and images were acquired with a LSM780 confocal microscope (Zeiss, San Diego, CA). For thick sections, the specimens were cleared overnight with Benzyl Alcohol/Benzyl Benzoate (1:2) solution (Sigma). 3D reconstruction of bone marrow was achieved by Z stack of tiled images of femoral bone marrow with a Zeiss LSM780 confocal microscope.

## Quantification of regressed sinusoids

We defined regressed sinusoids according to the criteria used in a previous publication (*Hooper et al., 2009*). We analyzed sinusoid morphology in thin optical sections through a segment of the femurs of mice. The sections were transverse sections through the longitudinal axis of the femurs, such that we observed cross-sections through most sinusoids. We first counted the total number of sinusoids in the section. Sinusoids were identified in these sections based on vessel morphology and bright VE-cadherin staining (VE-cadherin staining was dimmer in arterioles and capillaries). We then counted the number of regressed sinusoids in the same sections to arrive at the percentage of regressed sinusoids. Regressed sinusoids (*Figure 7D*) were distinguished from non-regressed sinusoids (*Figure 7C*) by being larger in diameter and having few hematopoietic cells around them.

## Evans blue extravasation assay

This assay was modified from a published method (*Radu and Chernoff, 2013*). Mice were retro-orbitally injected with 200 µl of 0.5% Evans blue in PBS and sacrificed 15 min later. Femurs and spleens were collected, crushed, and then Evans blue in these tissues was eluted in a set volume of PBS. After a brief centrifugation, the concentration of Evans blue in the supernatant was measured on a Nanodrop spectrophotometer (Thermo Scientific, Waltham, MA) at a wavelength of 610 nm. Femurs and spleens from mice without Evans blue injection were used as negative controls and blanks.

## Live imaging of calvarial bone marrow

Mice were anaesthetized by i.p. injection of ketamine/xylazine. Before imaging, the mice received a retro-orbital injection of 100 µl PBS solution containing 10 µg Alexa Fluor 660 conjugated anti-VE-cadherin antibody (BV13, eBiosciences) and 100 µg Dextran-FITC (70-kDa, Sigma). Then, the mouse was placed on a heated stage with the skull positioned under the objective using a stereotaxic device. Dextran-FITC fluorescence and autofluorescence from bone collagen were captured using two-photon imaging while Alexa Fluor 660-anti-VE-cadherin fluorescence was captured using confocal imaging on the same LSM780 microscope (Zeiss). An approximately 4 × 6 mm area of the calvarium encompassing most of the parasagittal bone marrow cavities within the left and right frontal bones was scanned in each imaging section.

## Scanning electron microscopy of bone marrow vasculature

Euthanized mice were pre-fixed by vascular perfusion via the left ventricle for 10 min with a solution containing 2% glutaraldehyde, 2% paraformaldehyde, and 0.1 M cacodylate buffer at pH 7.3. A 1 ml syringe fitted with a 23-gauge needle (BD Biosciences) containing ice-cold PBS was inserted into the growth plate and the then entire marrow plug was gently flushed from the marrow cavity. The marrow plugs were post-fixed in 2.5% glutaraldehyde overnight. They were partially dehydrated in ethanol, fractured in liquid nitrogen, rehydrated, and then fixed in 1% osmium tetroxide for another 2 hr. After full dehydration using a graded series of ethanol concentrations followed by hexamethyldisilazane, the specimens were coated with sliver. Two to three specimens per mouse were randomly chosen and examined on a Zeiss Sigma VP FE-SEM at 5–10 mkV at the UT Southwestern Electron Microscopy Core Facility.

## Peptide

Cavstratin, a fusion peptide of the putative scaffolding domain of caveolin-1 (amino acids 82–101: DGIWKASFTTFTVTKYWFYR) and the antennapedia internalization sequence (RQIKIWFQNRRMKWKK), was synthesized as previously described (*Gratton et al., 2003*) at the UT Southwestern Protein Chemistry Technology Center. Peptides were dissolved initially in DMSO and diluted 1000-fold in sterile PBS before in vivo administration (2.5 mg/kg per mouse).

## Acknowledgements

SJM is a Howard Hughes Medical Institute (HHMI) Investigator, the Mary McDermott Cook Chair in Pediatric Genetics, the director of the Hamon Laboratory for Stem Cells and Cancer, and a Cancer Prevention and Research Institute of Texas Scholar. This work was supported by the NIH NHLBI (HL097760). BOZ was supported by a fellowship from the Leukemia and Lymphoma Society. LD was supported by a Helen Hay Whitney Foundation Fellowship and by HHMI. We thank T Sanders and E Hughes at the UM transgenic core for helping to generate *Angpt1*$^{GFP}$ and *Angpt1*$^{fl}$ mice. We thank the UT Southwestern Protein Chemistry Technology Center for helping to synthesize the Cavtratin peptide. This work was initiated in the Life Sciences Institute at the University of Michigan then completed at Children's Research Institute at UT Southwestern.

## Additional information

### Competing interests

SJM: Senior editor, *eLife*. The other authors declare that no competing interests exist.

### Funding

| Funder | Grant reference | Author |
| --- | --- | --- |
| National Heart, Lung, and Blood Institute (NHBLI) | HL097760 | Sean J Morrison |
| Howard Hughes Medical Institute (HHMI) | Investigator | Sean J Morrison |
| Leukemia and Lymphoma Society (LLS) | fellowship | Bo O Zhou |
| Helen Hay Whitney Foundation (HHWF) | | Lei Ding |
| Howard Hughes Medical Institute (HHMI) | | Lei Ding |

The funders had no role in study design, data collection and interpretation, or the decision to submit the work for publication.

### Author contributions

BOZ, Acquisition of data, Analysis and interpretation of data, Drafting or revising the article; LD, Conception and design, Acquisition of data, Analysis and interpretation of data, Drafting or revising the article; SJM, Conception and design, Analysis and interpretation of data, Drafting or revising the article

## Ethics

Animal experimentation: This study was performed in strict accordance with the recommendations in the Guide for the Care and Use of Laboratory Animals of the National Institutes of Health. All mice were housed at the Unit for Laboratory Animal Medicine at the University of Michigan or in the Animal Resource Center at the University of Texas Southwestern Medical Center. All protocols were approved by the University of Michigan Committee on the Use and Care Animals and by the University of Texas Southwestern Institutional Animal Care and Use Committee (protocol 2011-0104).

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
