## [Decision Letter]

Thank you for sending your work entitled “Hematopoietic stem and progenitor cells regulate the regeneration of their niche by secreting Angiopoietin-1” for consideration at *eLife*. Your article has been favorably evaluated by Janet Rossant (Senior editor), a Reviewing editor, and 4 reviewers.

The Reviewing editor and the reviewers discussed their comments before we reached this decision, and the Reviewing editor has assembled the following comments to help you prepare a revised submission.

As you can see, all of your reviewers felt the work has merit and is in principle suited for *eLife*. In particular, the reviewers commented favorably on the rigorous quality of the study and the clarity of the writing. However, each of the reviewers raised some concerns, which you will want to address in revising the work. Here, I point to reviewer #1's major points, as resolving them does seem to be important to make the conclusions that you do. In fact, as pointed out by both reviewers 1 and 2, most effects do seem to be mediated by LepR+ stromal cells, rather than HSCs/progenitors. To unequivocally resolve this point, it seems reasonable to address whether the Ang1 knockout affects stromal cells in the bone marrow directly or indirectly, and in particular whether any of the cellular changes in the Mx1-Cre Ang1fl/fl bone marrow, when transplanted into LepR+ stromal Ang1 knockout recipients can promote HSC and vasculature regeneration that is HSC/HSC progeny independent. As pointed out by reviewers 1 and 3, it is important to determine whether Ang1, Ang2 or Tie2 expression change upon irradiation and upon transplantation during recovery. As the reviewers suggest, a transplantation experiment with just HSCs and their downstream lineages is needed to assess whether these proteins from HSCs/progenitors have an effect on their niche or whether the effects come solely from LepR+ stromal cells. Reviewer 3 also notes the sometimes opposing roles of Ang1 and Ang2, making this all the more important to investigate and resolve.

Reviewer 3 also raises some valid points regarding regression of vessels. Again, along the lines of reviewer 1 with regards to concern, the reviewer questions whether the loss of Ang-1 itself has a direct effect on BM recovery or whether the effect on BM recovery is secondary to vascular leakiness. The reviewer suggests that introducing exogenous Ang-1 into WT irradiated recipient, or injecting an unrelated anti-permeability agent in Mx-1→ Lepr irradiated recipients would help resolve this issue. The experiment seems straightforward and useful to solidify the claims made.

Finally, reviewer 4 raises two important issues regarding the claim that the Ang1/Tie2 axis has a role in regeneration of vasculature after irradiation. In particular, he/she raises 3 simple but important points, which merit addressing in your revised manuscript. The first issue, that of % of cells expressing Ang1 seems straightforward to correct. The issue of whether you've completely knocked out Ang1 in the osteoblasts is critical to resolve. The issue of the vasculature is one that has haunted all of your reviewers in different ways, and here it seems essential to shore up the additional issues raised by reviewer 4 in this regard.

If you can satisfactorily address the issues delineated above, *eLife* would welcome revised text and figures (and possibly a revised title, pending the outcome of the study).

Reviewer 1:

The manuscript by Zhou et al. examined the cellular source and functional importance of a secreted molecule Angiopoietin-1 (Ang1) in the HSC niche. Previous studies have identified HSCs and the perivascular stromal cells as potential source(s) for Ang1. Here, Bo et al. took one step forward and systematically knocked out Ang1 from a wide variety of cell types in the bone marrow to examine its function during homeostasis and upon irradiation. They demonstrated that although Ang1 is dispensable for HSC maintenance under steady condition, Ang1 delays HSC regeneration and bone marrow vasculature recovery after irradiation. Overall, this study represents a comprehensive analysis of Ang1's function in the HSC niche. The analysis is rigorous, and the manuscript is well written. How niche recovery is regulated after injury is an important topic which remains poorly understood. This study by Zhou et al. provides an interesting example, and will be of great interest to the broad readership of *eLife*. However, a few major points, if unresolved, can influence the authors' conclusions and interpretations.

The title states that “Hematopoietic stem and progenitor cells regulate the regeneration of their niche by secreting Angiopoietin-1”, which strongly implies that HSCs play a major and active role repairing their niche. However, from the presented data, most effects seem to be mediated by LepR+ stromal cells, while the evidence supporting the role of HSCs/progenitors is fairly weak. For example, transplanting wild-type bone marrow cells into Lepr-cre; Ang1fl/gfp recipients accelerate the regeneration of HSC and bone marrow vasculature, but transplanting Ang1 knockout bone marrow cells into irradiated wild type recipients does not have an effect. Although transplanting Ang1 bone marrow cells into Lepr-cre; Ang1fl/gfp recipients seem to have a greater effect, this data by itself does not unequivocally prove that the difference is mediated by HSCs and their progenitors due to the following reasons:

A) The expression of Mx1 promoter is fairly complex and dynamics, which includes HSCs, progenitors, and many different stroma cells in the bone marrow. In a bone marrow transplant, in addition to HSCs and progenitors, a wide variety of other cell types in the bone marrow are also transplanted together into the recipients. While many of these cells do not sustain in the recipient for long-term, short-term effects from these transplanted mesenchymal cells cannot be excluded. Does Ang1 knockout affect additional stroma cells in the bone marrow directly or indirectly? Can any of these cellular changes in the Mx1-Cre Ang1fl/fl bone marrow, when transplanted into recipients that has Ang1 knockout in the LepR+ stroma, promote HSC and vasculature regeneration that is HSC/HSC progeny independent? These possibilities cannot be formally ruled out with the experimental data presented.

B) Although the authors did a thorough analysis regarding Ang1 expression during steady state, does Ang1 expression change upon irradiation and upon transplantation (i.e., is it persistent in HSC lineages and the LepR+ cells? Do other cells which normally do not express high levels of Ang1 start to express Ang1? )

These issues will be difficult to resolve with merely negative data. Hence, the authors should do a transplantation experiment with just HSCs and their downstream lineages (e.g., isolate hematopoietic lineages from the Mx1-Cre; Ang1fl/fl mice and transplant into recipient mice). This will be an unambiguous support that Ang1 from Hematopoietic stem and progenitor cells truly has an effect on their niche. Otherwise it will be the HSC niche itself, and specifically the LepR+ cells, mediate niche recovery, with minor help from HSCs at the very best. This will still be an interesting discovery, but the authors will need to revise their Title and manuscript accordingly.

Reviewer 2:

Angiopoietin-1/Tie2 signaling has been shown to regulate adult hematopoiesis and hematopoietic recovery. Contradictory reports have been published on the source of Ang-1 in the hematopoietic stem cell niche and the role of osteoblasts in the niche. In this manuscript, the authors systematically evaluate Ang-1 expression in various BM HSC niche compartments by using state-of-art lineage analysis and excellent quality imaging. The role of Ang-1 in steady state hematopoiesis and HSC regeneration is studied by using various genetic models. It is quite clear from the BM transplantation experiments that Ang-1 can decrease hematopoietic recovery and deletion of Ang-1 improves the BM sinusoidal recovery. Ang-1 from LepR-expressing stromal cells seems to be a key player in regulating the niche regeneration.

This is an interesting study that settles a paradox in the field. The data are very convincing and thorough. Publication is recommended with high priority.

1) Col2.3-Cre;Ang-1fl/fl mice had normal WBC counts and normal HSC frequencies. However, in the transplanted mice, the T cell levels seemed to be affected during recovery. Could the authors comment on CLP levels in these mice and about their thymus?

2) The authors show that Ang-1 from endothelial or hematopoietic cells is not required for hematopoiesis by generating Tie2-cre;Ang-1fl/fl mice. Does it affect the angioblast differentiation? How does the endothelium look like? It would be good to check the deletion efficiency in these mice.

3) After irradiation, hematopoietic regeneration was accelerated in LepR+cell specific Ang-1 deleted mice at day 8 and day 12, although the CD150+CD48- HSCs frequency was not affected in Mx-1cre; ang1 mice. Could the authors comment on ST-HSCs?

4) Technical question: Methycellulose medum *M3434 may not be optimized for detecting CFU-Mk and it could be difficult to distinguish based on morphology. Did the authors stain the CFU-Mk or simply based on morphology?

Reviewer 3:

In the manuscript by Zhou et al. entitled “Hematopoietic stem and progenitor cells regulate the regeneration of their niche by secreting Angiopoietin-1”, the authors sought to determine the role of Ang-1 in the BM niche during irradiation recovery. The authors describe in detail that while Ang-1 is broadly expressed in early hematopoietic progenitors, that neither global loss of Ang-1, nor Ang-1 deletions from various compartments of the BM using different conditional knockouts affected hematopoiesis. Interestingly, deletion of Ang-1 from hematopoietic cells and LepR+ stromal cells accelerated vascular and hematopoietic recovery after radiation. While the work presented here is quite extensive and emphasizes the contribution of the vascular niche integrity during BM recovery, some revisions remain to be addressed in order for it to appeal to the readers of *eLife*:

1) What is the expression pattern change of Ang-1, Ang-2 and Tie2 in the BM after irradiation and during recovery? Even though Figure 2 indicates that hematopoietic cells and LepR+ cells are the major contributor for secreted Ang-1 during homeostasis, the absence of these population during injury can affect the dynamic of Ang/Tie2 signaling. Furthermore, Ang-1 and Ang-2 often have opposing effects on the vasculature so it would be important to see what role Ang-2 plays during recovery.

2) Can the authors more clearly define the parameters in which they measured the regressed vasculature in Figure 5 as this was not described in the Methods section? This is important because the vasculature is usually defined not just by the number of cells (such as % VE-Cadherin+ cells by FACS as in Figure 5–figure supplement 1) but by the vessel diameter, size, branch points etc. How do you define the regression of vessels?

3) Perhaps the most interesting but the most intriguing part of the paper is the discrepancy between the timeline of vessel regression (in Figure 5 which inversely correlated with the LSK recovery in Figure 4) and vascular permeability in Figure 6. It seems that vascular permeability was the most different on day 28 in Mx-1→ Lepr, AFTER vessel regression has caught up and normalized with concomitant LSK recovery. It is thus not clear to this reviewer whether the loss of Ang-1 itself has a direct effect on BM recovery or whether the effect on BM recovery is secondary to vascular leakiness. The authors stated in their Discussion that: “Together, the data indicate that Ang-1 produced by LepR+ stromal cells and hematopoietic cells promotes vascular integrity in regenerating blood vessels in the bone marrow at the cost of slowing the regeneration of HSC niches and hematopoiesis.” Please clarify and elaborate whether there might be a causal relationship between vascular leakiness and BM recovery? And if so, how can vascular leakiness promote BM recovery? The authors can also more definitively address this issue by introducing exogenous Ang-1 into WT irradiated recipient, or injecting an unrelated anti-permeability agent (such as those used in [21] Cancer Cell 4:31) in Mx-1→ Lepr irradiated recipients to see if BM recovery can be delayed.

Reviewer 4:

The manuscript by Zhou et al. look at hematopoiesis and bone marrow vascular affect in mice deleted for Ang1 in various populations of bone marrow cells. They conclude that the Ang1/Tie2 axis has a role in regeneration of vasculature after irradiation. There are 2 major issues regarding the observations. The effects observed by the authors are modest at best, and it is not entirely clear whether other cell populations other than those examined by the authors are contributing to vascular regeneration. Specific comments follow.

1) Statement that 72% of Kit+ cells but only Kit- cells are Ang1+ is misleading since there are many more Kit- cells than Kit+ cells.

2) As shown by the Lpr-cre experiments, homozygous deletion of two floxed alleles is difficult. How efficient was the deletion by Col2.3-cre, Osx-cre and Ubx-crER? Are there residual osteoblasts that have a normal allele?

3) The affects on vasculature are rather modest. Is this because Ang1 expression by cells other than the ones targeted by the various Cre vectors, incomplete deletion, or simply a modest role fo Ang1/Tie2 in blood vessel regeneration?

---

## [Author Response]

Reviewer 1:

*The title states that “Hematopoietic stem and progenitor cells regulate the regeneration of their niche by secreting Angiopoietin-1”, which strongly implies that HSCs play a major and active role repairing their niche. However, from the presented data, most effects seem to be mediated by LepR+ stromal cells, while the evidence supporting the role of HSCs/progenitors is fairly weak. For example, transplanting wild-type bone marrow cells into Lepr-cre; Ang1fl/gfp recipients accelerate the regeneration of HSC and bone marrow vasculature, but transplanting Ang1 knockout bone marrow cells into irradiated wild type recipients does not have an effect. Although transplanting Ang1 bone marrow cells into Lepr-cre; Ang1fl/gfp recipients seem to have a greater effect, this data by itself does not unequivocally prove that the difference is mediated by HSCs and their progenitors due to the following reasons*:

*A) The expression of Mx1 promoter is fairly complex and dynamics, which includes HSCs, progenitors, and many different stroma cells in the bone marrow. In a bone marrow transplant, in addition to HSCs and progenitors, a wide variety of other cell types in the bone marrow are also transplanted together into the recipients. While many of these cells do not sustain in the recipient for long-term, short-term effects from these transplanted mesenchymal cells cannot be excluded. Does Ang1 knockout affect additional stroma cells in the bone marrow directly or indirectly? Can any of these cellular changes in the Mx1-Cre Ang1fl/fl bone marrow, when transplanted into recipients that has Ang1 knockout in the LepR+ stroma, promote HSC and vasculature regeneration that is HSC/HSC progeny independent? These possibilities cannot be formally ruled out with the experimental data presented*.

The reviewer is correct that Angpt1 synthesized by LepR^+^ stromal cells appears to have a greater effect on the phenotypes we studied as compared to Angpt1 synthesized by hematopoietic cells. This is likely because in the early days (day 8 and 12) after irradiation and bone marrow transplantation, hematopoietic stem and progenitor cells are more rare than LepR^+^ stromal cells. However, *Lepr*^*cre*^*; Angpt1*^*GFP/fl*^ mice transplanted with *Angpt1* deficient bone marrow cells always had significantly faster vascular and bone marrow regeneration than *Lepr*^*cre*^*; Ang1*^*GFP/fl*^ mice transplanted with control bone marrow cells (Figures 6 and 7) as well as significantly greater vessel leakiness in the bone marrow (Figure 8). Therefore, the data are consistent with our conclusions.

*Mx1*-Cre does recombine in a variety of stromal cells other than hematopoietic cells in the bone marrow, including endothelial cells, perivascular stromal cells and osteoblasts. Conditional deletion of *Angpt1* by *Mx1*-Cre, or *Lepr*-Cre, or both, had no significant effects in normal mice on the numbers of VE-Cadherin^+^ endothelial cells or LepR^+^ perivascular cells (Figure 7), vascular integrity (Figure 8), hematopoiesis (Figures 3 and 6), or the morphology of the vasculature in the bone marrow (Figure 7—figure supplement 1). We are therefore unable to find any evidence that conditional *Angpt1* deletion in normal mice affects the frequency or quality of stromal cells in the bone marrow.

In the transplantation experiments performed using *Mx1-cre; Angpt1*^*fl/fl*^ bone marrow cells in the original manuscript it is unlikely that they contained significant numbers of stromal cells because the bone marrow cells were obtained without enzymatic dissociation. Enzymatic dissociation is required to obtain significant numbers of stromal cells from the bone marrow (Cell Stem Cell 15:154). Without enzymatic dissociation, the frequency of VE-Cadherin^+^ endothelial cells or LepR^+^ perivascular cells was less than one cell per million whole bone marrow cells (data not shown). Consistent with this, no CFU-F colonies could be detected when one million non-enzymatically dissociated bone marrow cells were cultured for 2 weeks (data not shown).

B) Although the authors did a thorough analysis regarding Ang1 expression during steady state, does Ang1 expression change upon irradiation and upon transplantation (i.e., is it persistent in HSC lineages and the LepR+ cells? Do other cells which normally do not express high levels of Ang1 start to express Ang1? )

We have not detected any changes in *Angpt1* expression after irradiation and transplantation. We have added new data to the manuscript in which we analyzed *Angpt1*-GFP expression in the bone marrow at days 8, 12, 16, and 28 after irradiation and bone marrow transplantation. Among hematopoietic cells, *Angpt1*-GFP expression was still highly restricted to c-kit^+^ hematopoietic progenitors and megakaryocytes at all time points analyzed after irradiation (Figure 6—figure supplement 2). It should be noted that at 8 days after irradiation no *Angpt1*-GFP^+^ hematopoietic cells were detected, consistent with the depletion of c-kit^+^ hematopoietic progenitors and megakaryocytes during the early phase of recovery from irradiation (Figure 6—figure supplement 2). Among stromal cells, *Angpt1*-GFP expression was still highly restricted to LepR^+^ perivascular stromal cells (Figure 6—figure supplement 2 and 2C). Even after irradiation we did not detect *Angpt1*-GFP expression by endothelial cells or osteoblasts (Figure 6—figure supplement 2). Therefore, we did not observe any changes in *Angpt1* expression after irradiation and transplantation.

*These issues will be difficult to resolve with merely negative data. Hence, the authors should do a transplantation experiment with just HSCs and their downstream lineages (e.g., isolate hematopoietic lineages from the Mx1-Cre; Ang1fl/fl mice and transplant into recipient mice). This will be an unambiguous support that Ang1 from Hematopoietic stem and progenitor cells truly has an effect on their niche. Otherwise it will be the HSC niche itself, and specifically the LepR+ cells, mediate niche recovery, with minor help from HSCs at the very best. This will still be an interesting discovery, but the authors will need to revise their Title and manuscript accordingly*.

We have performed two new experiments to address this issue directly. First, we transplanted 4000 LSK (Lineage^-^Sca-1^+^c-kit^+^) cells from control and *Mx1-cre; Angpt1*^*fl/fl*^ mice into *Lepr*^*cre*^*; Angpt1*^*GFP/fl*^ mice to test the effects of hematopoietic progenitors uncontaminated by stromal cells on vascular and hematopoietic regeneration after irradiation. We found that the mice transplanted with *Angpt1* deficient LSK cells had significantly higher bone marrow cellularity (Figure 6), LSK cell numbers (Figure 6), and better vascular morphology in the bone marrow (Figure 7) than mice transplanted with control LSK cells at 14 days after irradiation. The mice transplanted with *Angpt1* deficient, but not control, LSK cells also exhibited vascular leakage in bone marrow at 28 days after irradiation (Figure 8). These data demonstrate that *Angpt1* expression by hematopoietic cells regulates hematopoietic and vascular recovery after irradiation.

The only hematopoietic cells other than c-kit^+^ hematopoietic stem and progenitor cells that express Angpt1 are megakaryocytes (Figure 1). To test whether Angpt1 expression by megakaryocytes contributes to the regulation of hematopoietic and vascular recovery we conditionally deleted *Angpt1* from megakaryocyte lineage cells using *Pf4-*Cre and transplanted whole bone marrow cells from control and *Pf4-cre; Angpt1*^*fl/fl*^ mice into *Lepr*^*cre*^*; Angpt1*^*GFP/fl*^ recipients. We did not detect any significant differences in hematopoietic or vascular recovery between mice transplanted with control versus *Pf4-cre; Ang1*^*fl/fl*^ bone marrow (Figures 6, 7 and 8). These data suggest that Angpt1 expression by megakaryocyte lineage cells has little effect on hematopoietic and vascular recovery after irradiation. Together, our data demonstrate that Angpt1 expression by hematopoietic stem and progenitor cells regulates hematopoietic and vascular recovery after irradiation, along with Angpt1 expressed by LepR^+^ stromal cells.

Reviewer 2:

*1) Col2.3-Cre;Ang-1fl/fl mice had normal WBC counts and normal HSC frequencies. However, in the transplanted mice, the T cell levels seemed to be affected during recovery*. *Could the authors comment on CLP levels in these mice and about their thymus?*

We did not detect significant differences between control and *Col2.3-cre; Angpt1*^*fl/fl*^ mice with respect to CLP numbers in the bone marrow, thymus cellularity, or frequencies of CD4^+^ and/or CD8^+^ T cells in the thymus (Figure 3—figure supplement 2). The reduced T cell reconstitution from *Col2.3-cre; Angpt1*^*fl/fl*^ bone marrow was modest and transient as the difference was not statistically significant at 16 weeks after transplantation (Figure 3). This is consistent with our observation that osteoblasts do not express *Angpt1*.

*2) The authors show that Ang-1 from endothelial or hematopoietic cells is not required for hematopoiesis by generating Tie2-cre;Ang-1fl/fl mice. Does it affect the angioblast differentiation? How does the endothelium look like? It would be good to check the deletion efficiency in these mice*.

We looked at the bone marrow endothelium by whole-mount VE-Cadherin staining in control and *Tie2-cre; Angpt1*^*fl/fl*^ mice at two months of age and did not detect any differences in the morphology or density of the vasculature in the bone marrow (see new Figure 5). This is consistent with a prior study that found ubiquitous deletion of *Angpt1* after E13.5 did not affect vascular morphology or function in normal uninjured mice (J. Clin. Invest. 121:2278). We tested the deletion efficiency of *Angpt1*^*fl*^ in VE-Cadherin^+^CD45^-^Ter119^-^ endothelial cells and CD45^+^Ter119^+^ hematopoietic cells obtained from *Tie2-cre; Angpt1*^*fl/fl*^ mice. The deletion efficiencies in both cell populations were approximately 97% (see new Figure 5). We do not know how to identify angioblasts or to test their function so would need more direction from the reviewer if the foregoing data are not sufficient.

*3) After irradiation, hematopoietic regeneration was accelerated in LepR+cell specific Ang-1 deleted mice at day 8 and day 12*, *although the CD150+CD48- HSCs frequency was not affected in Mx-1cre; ang1 mice. Could the authors comment on ST-HSCs?*

The frequency of CD150^-^CD48^-^LSK ST-HSCs (Figure 3—figure supplement 2) or LSK cells (Figure 6) was also not significantly different between *Mx1-cre; Angpt1*^*fl/fl*^ and control bone marrow in normal mice.

4) Technical question: Methycellulose medum *M3434 may not be optimized for detecting CFU-Mk and it could be difficult to distinguish based on morphology. Did the authors stain the CFU-Mk or simply based on morphology?

We supplemented methylcellulose medium M3434 with TPO (10 ng/ml) and FLT3 (10 ng/ml) to optimize the detection of CFU-Mk. This allowed us to detect Mk colonies formed by single HSCs really well. The CFU-Mk colonies were identified based on morphology, based on the presence of morphologically distinct, large megakaryocytes.

Reviewer 3:

*1) What is the expression pattern change of Ang-1, Ang-2 and Tie2 in the BM after irradiation and during recovery? Even though*
Figure 2
*indicates that hematopoietic cells and LepR+ cells are the major contributor for secreted Ang-1 during homeostasis, the absence of these population during injury can affect the dynamic of Ang/Tie2 signaling. Furthermore, Ang-1 and Ang-2 often have opposing effects on the vasculature so it would be important to see what role Ang-2 plays during recovery*.

As requested, we added new data showing the expression of *Angpt1*, *Angpt2* and Tie2 in the bone marrow after irradiation and bone marrow transplantation. The short answer is that we have not detected any significantly changes in the expression patterns of any of these genes in the bone marrow after irradiation and transplantation.

We have not detected any changes in *Angpt1* expression after irradiation and transplantation. We analyzed *Angpt1*-GFP expression in the bone marrow at days 8, 12, 16, and 28 after irradiation and bone marrow transplantation. Among hematopoietic cells, *Angpt1*-GFP expression was still highly restricted to c-kit^+^ hematopoietic progenitors and megakaryocytes at all time points analyzed after irradiation (Figure 6—figure supplement 2). It should be noted that at 8 days after irradiation no *Angpt1*-GFP^+^ hematopoietic cells were detected, consistent with the depletion of c-kit^+^ hematopoietic progenitors and megakaryocytes at early phases of recovery from irradiation (Figure 6—figure supplement 2). Among stromal cells, *Angpt1*-GFP expression was still highly restricted to LepR^+^ perivascular stromal cells (Figure 6—figure supplement 2). Even after irradiation we did not detect *Angpt1*-GFP expression by endothelial cells or osteoblasts (Figure 6—figure supplement 2).

Consistent with prior studies (Cancer Letters 328:18) we found that *Angpt2* was expressed mainly by endothelial cells in the bone marrow under normal conditions as well as after irradiation (Figure 6—figure supplement 2). Depending on the context, Angpt1 and Angpt2 may have either opposing or synergistic effects on the vasculature (Nat Rev Mol Cell Biol 10:165; J. Clin. Invest. 124:4320). While it would be interesting to assess the function of Angpt2 during recovery after irradiation we believe these studies are outside the scope of this manuscript as they would require us to obtain *Angpt2*^*flox*^ mice (which we do not have) and to redo all of our experiments in new genetic backgrounds. No matter how those experiments turn out, they would not affect any of the conclusions of the current manuscript.

In both normal adult bone marrow and at days 12-28 after irradiation and transplantation, Tie2 was expressed by most LSK hematopoietic stem/progenitors cells, most c-kit+ hematopoietic progenitors, and most endothelial cells (see new Figure 6—figure supplement 2). At day 8 after irradiation Tie2 was still expressed by most endothelial cells but there LSK cells and c-kit^+^ cells were extremely rare in the bone marrow. Tie2 was rarely expressed by LepR^+^ stromal cells or c-kit negative hematopoietic cells either in normal bone marrow or after irradiation (see new Figure 6—figure supplement 2).

*2) Can the authors more clearly define the parameters in which they measured the regressed vasculature in*
Figure 5
*as this was not described in the Methods section? This is important because the vasculature is usually defined not just by the number of cells (such as % VE-Cadherin+ cells by FACS as in Figure 5–figure supplement 1) but by the vessel diameter, size, branch points etc. How do you define the regression of vessels?*

We defined regressed sinusoids according to the criteria used in a previous publication from Shahin Rafii’s lab (Cell Stem Cell 4:263). Regressed sinusoids are dilated and have fewer surrounding hematopoietic cells as compared to normal sinusoids. We analyzed sinusoid morphology in thin optical sections through a segment of the femurs of mice. The sections were transverse sections through the longitudinal axis of the femurs, such that we would observe cross-sections through most sinusoids. In these thin optical sections, we first counted the total number of sinusoids in the section. Sinusoids were identified in these sections based on vessel morphology and bright VE-cadherin staining (VE-cadherin staining was dimmer in arterioles and capillaries). We then counted the number of regressed sinusoids in the same sections to arrive at the percentage of regressed sinusoids, as shown in Figure 7. Regressed sinusoids were distinguished from non-regressed sinusoids by being larger in diameter and having few hematopoietic cells around them. This is illustrated in a new Figure 7 (showing normal sinusoids) versus 7D (showing regressed sinusoids) that we have added to the manuscript. In our experience there is regional recovery of the bone marrow, where some regions recover normal sinusoid morphology and hematopoiesis before others. Statistics are based upon analyses of three sections per mouse and at least three mice per time and treatment. We have added a discussion of these issues to the Methods.

*3) Perhaps the most interesting but the most intriguing part of the paper is the discrepancy between the timeline of vessel regression (in*
Figure 5
*which inversely correlated with the LSK recovery in*
Figure 4*) and vascular permeability in*
Figure 6*. It seems that vascular permeability was the most different on day 28 in Mx-1→ Lepr, AFTER vessel regression has caught up and normalized with concomitant LSK recovery. It is thus not clear to this reviewer whether the loss of Ang-1 itself has a direct effect on BM recovery or whether the effect on BM recovery is secondary to vascular leakiness. The authors stated in their Discussion that: “Together, the data indicate that Ang-1 produced by LepR+ stromal cells and hematopoietic cells promotes vascular integrity in regenerating blood vessels in the bone marrow at the cost of slowing the regeneration of HSC niches and hematopoiesis.” Please clarify and elaborate whether there might be a causal relationship between vascular leakiness and BM recovery? And if so, how can vascular leakiness promote BM recovery? The authors can also more definitively address this issue by introducing exogenous Ang-1 into WT irradiated recipient, or injecting an unrelated anti-permeability agent (such as those used in*
[21]
*Cancer Cell 4:31) in Mx-1→ Lepr irradiated recipients to see if BM recovery can be delayed*.

To test whether there is a causal relationship between vascular leakiness and hematopoietic recovery we injected Cavtratin, the unrelated anti-permeability agent suggested by the reviewer (Cancer Cell 4:31), in control and Mx1→Lepr recipients from 7 to 13 days after irradiation then analyzed the mice. Based on both Dextran-FITC live-imaging and Evans blue extravasation, Cavtratin administration significantly reduced vascular leakiness in both control and Mx1→Lepr recipient mice (Figure 9). However, Cavtratin administration did not significantly affect the recovery of bone marrow cellularity or LSK cell numbers in the bone marrow of control or Mx1→Lepr recipients (Figure 9). Mx1→Lepr recipients continued to regenerate marrow cellularity and LSK cell numbers significantly faster than control mice, irrespective of Cavtratin treatment. These data demonstrate that the accelerated recovery of hematopoietic stem/progenitor cells and hematopoiesis in the absence of *Angpt1* is not caused by the increase in vascular leakiness. These appear to reflect independent effects of *Angpt1*.

Reviewer 4:

*The manuscript by Zhou et al. look at hematopoiesis and bone marrow vascular affect in mice deleted for Ang1 in various populations of bone marrow cells. They conclude that the Ang1/Tie2 axis has a role in regeneration of vasculature after irradiation. There are 2 major issues regarding the observations. The effects observed by the authors are modest at best, and it is not entirely clear whether other cell populations other than those examined by the authors are contributing to vascular regeneration. Specific comments follow*.

*1) Statement that 72% of Kit+ cells but only Kit- cells are Ang1+ is misleading since there are many more Kit- cells than Kit+ cells*.

To clearly show the relationship between c-kit and *Angpt1* expression we have included a new figure in the revised manuscript showing c-kit versus *Angpt1-*GFP expression among mechanically dissociated bone marrow cells (Figure 1—figure supplement 1). The data show that about 85% of *Angpt1*-GFP^+^ hematopoietic cells are c-kit^+^.

2) As shown by the Lpr-cre experiments, homozygous deletion of two floxed alleles is difficult. How efficient was the deletion by Col2.3-cre, Osx-cre and Ubx-crER? Are there residual osteoblasts that have a normal allele?

It is important to note that we have not been able to detect any *Angpt1* expression by osteoblasts (see Figure 2), consistent with the lack of any functional phenotype upon conditionally deleting in these cells. Nonetheless, as requested we have added new data to the manuscript measuring the deletion efficiency of *Angpt1*^*fl*^ by *Col2.3*-Cre, *Osx*-Cre and *UBC*-Cre/ER. The recombination efficiency of *Angpt1*^*fl*^ was measured by real-time PCR analysis of genomic DNA from flow cytometrically purified cells. The amplification of the *Angpt1*^*fl*^ allele in mutant mice was compared to the amplification of the same product from *Angpt1*^*fl/fl*^ control mice. An unrelated genomic locus was amplified in parallel to normalize DNA content among samples.

*Col2.3*-Cre deleted 94±3.0% of *Angpt1*^*fl*^ alleles in *Col2.3-GFP*^+^ osteoblasts from *Col2.3-cre; Angpt1*^*fl/fl*^*; Col2.3-GFP* mice (see new Figure 3—figure supplement 2). *Osx*-Cre deleted 93±3.0% of *Angpt1*^*fl*^ alleles in CD45^-^Ter119^-^CD31^-^PDGFRα^+^CD105^+^ osteoprogenitors from *Osx-cre; Angpt1*^*fl/fl*^ mice (see new Figure 4). *UBC*-Cre/ER deleted 95±2.2% of *Angpt1*^*fl*^ alleles in LSK cells from *Ubc-creER; Angpt1*^*fl/fl*^ mice administered tamoxifen-containing chow for at least a month (see new Figure 3—figure supplement 2).

3) The effects on vasculature are rather modest. Is this because Ang1 expression by cells other than the ones targeted by the various Cre vectors, incomplete deletion, or simply a modest role fo Ang1/Tie2 in blood vessel regeneration?

There are a number of possibilities. First, there could be some compensation by other Angiopoietin family members. Second, our approach of conditionally deleting *Angpt1* from specific cell types has the advantage of identifying the physiologically important sources of Angpt1 in the bone marrow but the disadvantage that secondary sources could partially rescue the phenotype when we delete from the main sources (hematopoietic stem/progenitor cells and LepR^+^ stromal cells). Finally, it is possible that the function of Angpt1 is modest.

No matter which of the above turn out to be true, our manuscript will have made two important points. First, that the function of Angpt1 in the hematopoietic system is quite different from what was originally claimed in a highly cited paper (it is not an osteoblast specific promoter of HSC maintenance and quiescence). Second, our data make the conceptually important point that hematopoietic stem/progenitor cells regulate niche regeneration after injury. We believe this is the first clear evidence that normal HSCs regulate their niche. All prior studies have focused on mechanisms by which the niche regulates HSCs.